# The spike gene is a major determinant for the SARS-CoV-2 Omicron-BA.1 phenotype

Variant of concern (VOC) Omicron-BA.1 has achieved global predominance in early 2022. Therefore, surveillance and comprehensive characterization of Omicron-BA.1 in advanced primary cell culture systems and animal models are urgently needed. Here, we characterize Omicron-BA.1 and recombinant Omicron-BA.1 spike gene mutants in comparison with VOC Delta in well-differentiated primary human nasal and bronchial epithelial cells in vitro, followed by in vivo fitness characterization in hamsters, ferrets and hACE2-expressing mice, and immunized hACE2-mice. We demonstrate a spike-mediated enhancement of early replication of Omicron-BA.1 in nasal epithelial cultures, but limited replication in bronchial epithelial cultures. In hamsters, Delta shows dominance over Omicron-BA.1, and in ferrets Omicron-BA.1 infection is abortive. In hACE2-knock-in mice, Delta and a Delta spike clone also show dominance over Omicron-BA.1 and an Omicron-BA.1 spike clone, respectively. Interestingly, in naïve K18-hACE2 mice, we observe Delta spike-mediated increased replication and pathogenicity and Omicron-BA.1 spike-mediated reduced replication and pathogenicity, suggesting that the spike gene is a major determinant of replication and pathogenicity. Finally, the Omicron-BA.1 spike clone is less well-controlled by mRNA-vaccination in K18-hACE2-mice and becomes more competitive compared to the progenitor and Delta spike clones, suggesting that spike gene-mediated immune evasion is another important factor that led to Omicron-BA.1 dominance.

On a global scale, SARS-CoV-2 evolution can be tracked by identifying independently emerging variants of concern (VOCs), with VOC Alpha, Delta, and Omicron dominating successively. Delta carries two deterministic mutations potentially leading to increased fitness: L452R, conferring immune escape[1], and P681R, conferring enhanced transmission[2]. Omicron-BA.1 holds in total up to 50 mutations, with 34 located in the spike (S) gene, 15 of which are within the receptor-binding domain (RBD)[3]. A defining Omicron-BA.1 mutation is ins214EPE, a three-amino acid insertion, whose role for viral fitness is still unknown. However, this VOC is characterized by its remarkable ability to evade neutralizing antibodies up to 40 times more efficiently than the ancestral SARS-CoV-2 and pre-Omicron variants[4,5]. In January 2022, the Omicron-BA.1 lineage became predominant in most countries worldwide[6] and has since then largely been replaced by the related Omicron-BA.2 and other Omicron sublineages. It remains elusive if the rapid spread of Omicron-BA.1 and the replacement of Delta is due to increased fitness and transmission, or if it is mainly based on its immune escape ability allowing efficient infection and transmission chains among double-vaccinated and even boosted individuals. The genetic determinants for the Omicron-BA.1 phenotype also remain largely undefined. With high prevalence of concurrent VOCs and reports of recombination events in communities[7], it is crucial to characterize differences in viral fitness and immune escape of emerging and prevailing VOCs in advanced cell culture and animal models[8].

While mouse models expressing human angiotensin-converting enzyme 2 (hACE2) and Syrian hamsters are highly susceptible for SARS-CoV-2 and show signs of severe disease, ferrets display subclinical infection despite efficient viral replication of SARS-CoV-2 in the

✉e-mail: marco.alves@vetsuisse.unibe.ch; charaf.benarafa@vetsuisse.unibe.ch; martin.beer@fli.de; volker.thiel@vetsuisse.unibe.ch

upper respiratory tract (URT)[9,10]. An experimental setup applying competitive infection and transmission experiments in different species has become a gold-standard method to investigate VOC fitness[9–12]. With this experimental approach, the fitness of SARS-CoV-2 VOCs can be analyzed in direct comparison at the nucleotide and variant level.

Here, we demonstrate a dominance of Delta over Alpha in ferrets, whereas in Syrian hamsters, Alpha dominated Delta. Moreover, we demonstrate that the advent of the Omicron VOC in the evolution of SARS-CoV-2 is a radical change from the incremental improvements in fitness observed in previous pandemic VOCs. Using a comprehensive experimental VOC competition approach against Delta, we demonstrate that the Omicron-BA.1 phenotype is characterized by (i) a reduced replication and transmission fitness in Syrian hamsters, (ii) a failure to replicate in ferrets, (iii) an accelerated growth in human epithelial cell cultures mimicking the upper respiratory tract, (iv) a reduced replication in precision-cut lung slices (PCLS) and primary human bronchial epithelial cultures, resembling conditions of the human lower respiratory tract, (v) a reduced replication fitness in naïve human ACE2 (hACE2) expressing knock-in (hACE2-KI) and transgenic (K18-hACE2) mice, and (vi) evidence of immune evasion in mRNA-vaccinated K18-hACE2 mice. Importantly, we show that the spike gene is a major determinant in the Omicron-BA.1 phenotype based on in vitro and in vivo experiments using recombinant SARS-CoV-2 clones differing only by the expression of the spike protein of the respective VOCs.

## Results

### Omicron-BA.1 spike enhances viral replication in the nasal but not bronchial epithelium

In order to assess the phenotypes of the VOCs Delta and Omicron-BA.1 and to evaluate the contribution of changes within the spike protein, we constructed a set of recombinant SARS-CoV-2 clones containing defined mutations in the spike gene (Fig. 1a). All constructs have an isogenic background based on the Wuhan-Hu-1 sequence and differ only in the spike gene, which was modified to contain lineage-defining spike gene mutations of the VOC Delta (SARS-CoV-2$^{S-Delta}$), VOC Omicron-BA.1 (SARS-CoV-2$^{S-Omicron}$), mutations of the Omicron-BA.1 spike N-terminal domain (NTD; SARS-CoV-2$^{NTD-Omicron}$), mutations of the Omicron-BA.1 spike receptor-binding domain (RBD; SARS-CoV-2$^{RBD-Omicron}$), or the mutations at and near the Omicron-BA.1 spike cleavage site region (CS; SARS-CoV-2$^{CS-Omicron}$). All recombinant viruses replicated but showed noticeable differences in plaque sizes (Fig. 1b). Compared to the index virus SARS-CoV-2$^{D614G}$ (recombinant SARS-CoV-2 based on Wuhan-Hu-1 with the spike change D614G[11]), the Delta isolate showed smaller plaques, while plaques of SARS-CoV-2$^{S-Delta}$ were considerably larger. Interestingly, SARS-CoV-2$^{S-Omicron}$ displayed small plaques that were indistinguishable from the isolate of Omicron-BA.1 (EPI_ISL_7062525), but plaques of Omicron-BA.1 spike subdomain clones (SARS-CoV-2$^{NTD-Omicron}$, SARS-CoV-2$^{CS-Omicron}$, SARS-CoV-2$^{RBD-Omicron}$) differed in size (Fig. 1b), indicating possible phenotypic differences of the Omicron-BA.1 spike subdomain clones compared to the Omicron-BA.1 isolate and the full-length Omicron-BA.1 spike clone.

Next, we infected well-differentiated primary human nasal and bronchial epithelial cell cultures (hNECs and hBECs, respectively) at 33 °C for hNECs and at 37 °C for hBECs (Fig. 1c, d). Delta and the corresponding spike construct SARS-CoV-2$^{S-Delta}$ replicated with similar kinetics as wild-type SARS-CoV-2$^{D614G}$ on both hNECs and hBECs, with SARS-CoV-2$^{S-Delta}$ reaching the highest apical titers at 72–96 h post-infection (hpi) (Fig. 1c). Strikingly, replication kinetics of Omicron-BA.1 and the corresponding spike clone SARS-CoV-2$^{S-Omicron}$ displayed accelerated growth within the first 48 hpi on hNECs. In contrast, on hBECs, Omicron-BA.1 and SARS-CoV-2$^{S-Omicron}$ did not show this early accelerated growth, and moreover, showed significantly reduced viral titers at later time points. This phenotype was confirmed by

competition assays on hNECs and hBECs using various combinations of viruses in the inoculum (Fig. 1e, f; Supplementary Fig. 1). On hNECs, the Omicron-BA.1 isolate and the corresponding SARS-CoV-2$^{S-Omicron}$ outcompeted SARS-CoV-2$^{D614G}$, the Delta isolate, and SARS-CoV-2$^{S-Delta}$ (Fig. 1d). In contrast, the dominance of the Omicron-BA.1 isolate, and SARS-CoV-2$^{S-Omicron}$ was reduced in hBECs (Fig. 1d). Finally, SARS-CoV-2$^{S-Delta}$ was dominant over SARS-CoV-2$^{S-Omicron}$ in an ex vivo distal precision-cut lung slices (PCLS) system (Fig. 1g).

Collectively, studies in vitro under conditions mimicking the native human upper respiratory tract epithelium (URT; hNECs at 33 °C) remarkably demonstrate that the Omicron-BA.1 spike gene (Omicron-BA.1 isolate and SARS-CoV-2$^{S-Omicron}$) confers accelerated virus replication and increased replicative fitness compared to pre-Omicron spike genes (SARS-CoV-2$^{D614G}$, a Delta isolate, and SARS-CoV-2$^{S-Delta}$). In contrast, under conditions resembling the human lower respiratory tract epithelium (LRT; hBECs, 37 °C; PCLS), the Omicron-BA.1 spike gene confers reduced virus replication.

### Replicative fitness and transmission of VOCs Alpha, Delta, and Omicron in Syrian hamsters

To evaluate individual VOC fitness advantages in direct competition with the precursing VOC, we then turned to animal models with natural susceptibility towards SARS-CoV-2: ferrets and Syrian hamsters. Donor animals were simultaneously co-inoculated with two VOCs at iso-titer and transmission to contact animals was investigated (Supplementary Fig. 2a, b).

We investigated fitness of the VOC Delta in competition with Alpha after intranasal co-inoculation of $10^{4.625}$ TCID$_{50}$ of an Alpha-Delta mixture at a 1.95:1 ratio (Supplementary Figs. 2a and 5). For all donor and contact hamsters, Alpha showed complete dominance in nasal washings and respiratory tissues (Supplementary Figs. 5 and 8). The animals showed signs of severe disease and 10 of 12 contact hamsters reached the humane endpoint, (defined by a 20% loss of the initial body weight) (Supplementary Figs. 3a, c and 5). While Alpha was clearly dominant in all animals, sera equally neutralized Alpha and Delta VOCs (VNT$_{100}$) (Supplementary Fig. 7a).

We also tested Delta vs Omicron-BA.1 (total $10^{4.5}$ TCID$_{50}$ at a 1:2.16 ratio) in hamsters (Fig. 2a, Supplementary Fig. 2a). Upon inoculation or contact, all animals lost body weight, however, only one animal was euthanized after reaching the humane endpoint for body weight loss (Fig. 2a, Supplementary Fig. 3b, d). Although starting with a clear advantage for Omicron-BA.1 in the inoculum, Delta was immediately prevalent in nasal washings of donor hamsters with up to $10^8$ genome copies per mL (GE/mL) (Fig. 2a). Remarkably, despite high genome loads of Delta, Omicron-BA.1 still replicated to $10^7$ GE/mL (Fig. 2a). Nevertheless, Delta was preferentially transmitted as seen in both nasal washings and organ samples of contact animals (Fig. 2a, Supplementary Fig. 4). In tissues of donor animals at 4 dpi, mainly Delta was found in the upper (URT) and lower respiratory tract (LRT) (Fig. 2b); with highest Delta loads in the nasal concha ($10^8$ GE/mL). However, Omicron-BA.1 was still present in the URT of each donor animal (up to $10^7$ GE/mL) (Fig. 2b). Consequently, the antibody response was mainly directed against Delta with neutralization up to a serum dilution of 1:1024, while Omicron-BA.1 was only barely neutralized (Fig. 2d). Together, we show that the Syrian hamster is highly susceptible also for the SARS-CoV-2 VOCs Alpha, Delta, and Omicron-BA.1. However, the Alpha VOC seems to be best replicating and transmitting in the Syrian hamster model associated with the highest fatality rates.

### Omicron-BA.1 fails to induce productive infection in ferrets

For further characterization of the different VOCs, we also inoculated ferrets, which are known to mirror human respiratory disease. First, animals were inoculated with a Delta isolate (Supplementary Fig. 2c). All animals remained clinically healthy and did not lose body weight

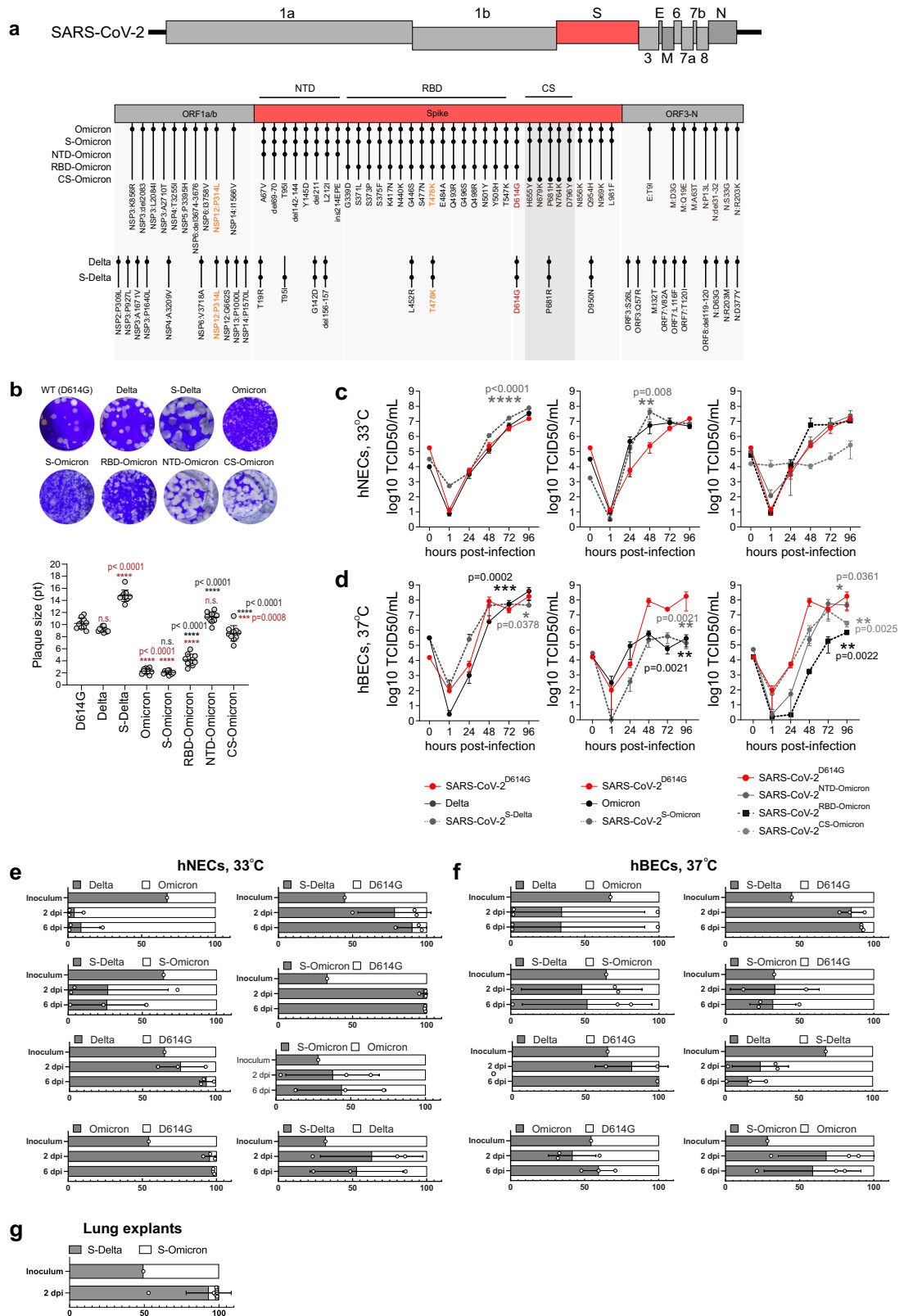

(Supplementary Fig. 3e). Viral shedding was confirmed for all donor animals by nasal washing for up to 10 days, with highest viral genome loads of up to >10[7] GE/mL at 5 and 6 dpi (Fig. 2f). Two out of three contact animals were infected by 6 days post contact (dpc) and viral loads in contacts reached up to 10[8] GE/mL (Fig. 2f). These results were also confirmed by serology (Fig. 2f).

Next, ferrets were co-inoculated with VOCs Alpha and Delta (1.33:1 ratio, 10[5] TCID$_{50}$ in total) (Supplementary Figs. 2b and 6). All ferrets remained clinically healthy throughout the animal experiment and did not lose body weight (Supplementary Fig. 3g). Within 3 days, Delta fully outcompeted Alpha in both replication and transmission (Supplementary Fig. 6), which is in line with the epidemiological situation

**Fig. 1 | Enhanced replication of Omicron-BA.1 in nasal but not bronchial epithelial cell cultures. a** Genome sequences were compared to the SARS-CoV-2$^{D614G}$ WT virus and lineage-defining mutations (LDM) are depicted. The D614G mutation is highlighted in red, while the mutations highlighted in orange are either present in both Delta and Omicron, or in Omicron, S-Omicron, RBD-Omicron, Delta, and S-Delta, but not in SARS-CoV-2$^{D614G}$. **b** The plaque sizes of viruses in 6-well plates 2 dpi. Sizes of 10 plaques/wells from one biological replicate were measured in Adobe Illustrator. Data are presented as mean+/−SD. Statistical significance was determined using ordinary one-way Anova and p-values were adjusted using Tukey's multiple-comparison test; *P < 0.05, **P < 0.01, ***P < 0.001, ****P < 0.0001. Statistical significance of the differences of each virus vs. SARS-CoV-2$^{D614G}$ WT virus are demarcated with the red asterisks, whereas the black asterisks indicate the comparison of the Omicron spike subdomain clones to Omicron. **c, d** Human nasal (NEC) (n = 3 donors) and bronchial epithelial cell (BEC) (n = 3) cultures were infected with $10^4$ TCID$_{50}$ of the SARS-CoV-2 variants from the apical side and

incubated at 33 °C (NECs) or 37 °C (BECs) for 1 h. Virus titers were assessed by TCID$_{50}$ assays on VeroE6/TMPRSS2 cells. The graph represents the titers obtained from three donors (mean+/−SD) from one biological replicate. Statistical significance was determined using two-way ANOVA and p-values were adjusted using Tukey's multiple-comparison test; *P < 0.05, **P < 0.01, ***P < 0.001, ****P < 0.0001. **e–g** NECs (n = 3), BECs (n = 3), or PCLS (n = 3) were infected with virus mixtures at a 1:1 ratio based on genome equivalents (GE) calculated by qPCR. Apical washes were collected at 2 and 6 dpi for the NECs and BECs and 2 dpi for PCLS. RNA was extracted from apical washes and sequenced on the MinION platform (Oxford Nanopore Technologies). Virus ratios were calculated for each donor based on the mean frequency of unique LDM mutations for each virus present in the mixture (for more details: Supplementary Fig. 1). Values shown represent the mean ratio/donor (circles) and the mean ratio/time point (bars) for each virus mixture (mean+/−SD). Each data point represents one biological replicate. Source data for Fig. 1 are provided as a Source Data file 1.Source Data file 1.

observed in humans. These observations were further supported by the matching serological data (Supplementary Fig. 7b).

We finally investigated the competition between Delta and Omicron-BA.1 in the ferret model by inoculating a $10^{4.75}$ TCID$_{50}$ Delta-Omicron mixture (1:1.43 ratio) (Fig. 2c, Supplementary Fig. 2B). All ferrets remained clinically healthy throughout the animal experiment and did not lose body weight (Supplementary Fig. 3h). Astonishingly, only Delta was detected in all nasal washings of the donor ferrets, starting at 1 dpi at levels of up to $10^7$ GE/mL (Fig. 2c). In addition, 5 out of 6 contact ferrets showed shedding of Delta (highest loads: $10^7$ GE/mL), starting at 2 dpc (Fig. 2c). The shedding interval lasted up to 12 days in the donor ferrets (Fig. 2c). Surprisingly, Omicron-BA.1 was not detected in any donor, hence only Delta was transmitted to contact ferrets (Fig. 2c). These results indicate a severe block of Omicron-BA.1 infection in ferrets. To confirm the unexpected observation, we inoculated ferrets with $10^{5.125}$ TCID50 of Omicron-BA.1 (Supplementary Fig. 2c). All ferrets remained clinically healthy throughout the study with no marked body weight changes (Supplementary Fig. 3f). Again, we detected neither shedding of Omicron-BA.1 in nasal washings, nor vRNA in the URT or LRT of ferrets euthanized at 6dpi (Fig. 2f). Serological analysis confirmed the RT-qPCR results by revealing lack of seroconversion at 21 dpi (Fig. 2f). Successful back-titration of the Omicron-BA.1 inocula for each experiment, whole-genome sequence confirmation by high-throughput sequencing and using the same virus stock for both the hamster and the ferret experiments, strongly suggest a complete replication block of VOC Omicron-BA.1 in ferrets. Therefore, the Delta variant seems to exhibit the top-level fitness in ferrets, and the vast changes in the Omicron-BA.1 sequence might be at the cost of broad host spectrum.

### Delta spike mutations drive enhanced fitness in hACE2 knock-in mice

To mitigate the impact of mismatched interactions between non-human ACE2 and the Omicron-BA.1 spike/RBD, knock-in mice expressing only human ACE2 (hACE2-KI) were inoculated intranasally with $10^{4.3}$ TCID$_{50}$/mouse of Omicron-BA.1 or Delta. Infection with Delta only caused body weight loss at 4dpi. Higher virus loads and titers in the URT, LRT, and olfactory bulb were found in mice inoculated with Delta compared to Omicron-BA.1 (Fig. 3a–c, Supplementary Fig. 9a). Higher virus loads in hACE2-KI mice infected with Delta were associated with a higher pathological score in the lungs, which showed multifocal, peribronchiolar inflammatory cuffs (Fig. 3d, e, Supplementary Information Table 2). In a competition setting, Delta also dominated over Omicron-BA.1 in the URT and LRT (Fig. 3f, Supplementary Fig. 9b). To determine the importance of the spike mutations, hACE2-KI mice were inoculated with an equivalent mixture of recombinant clones only differing by the spike sequence, SARS-CoV-2$^{S-Delta}$ and SARS-CoV-2$^{S-Omicron}$. As for the VOC isolates, SARS-CoV-2$^{S-Delta}$ fully dominated over SARS-CoV-2$^{S-Omicron}$ in the URT and LRT of hACE2-KI

mice indicating that the spike is of major importance for the phenotype of both VOCs and that spike mutations do not provide an advantage to Omicron-BA.1 over Delta in the hACE2-KI mouse model (Fig. 3g). These findings suggest that the intrinsic replicative properties of Omicron-BA.1 are unlikely to be the decisive factor for the observed replacement of Delta by Omicron-BA.1 in the human population.

### Omicron-BA.1 spike confers immune escape and reduced pathogenicity

Next, we investigated the impact of prior SARS-CoV-2-specific mRNA vaccination on the replicative fitness of Omicron-BA.1 and Delta (Fig. 4a). Since current mRNA vaccines use the spike as the sole viral antigen, we focused on the impact of the spike gene on immune evasion and replication. Further, to exclude any influence of mutations outside the spike genes of the Delta and the Omicron-BA.1 VOC on replication, immune evasion, or pathogenicity, we used the isogenic viruses SARS-CoV-2$^{S-Delta}$ and SARS-CoV-2$^{S-Omicron}$ that differ only in their spike gene for infection of vaccinated and naïve mice. Transgenic K18-hACE2 were immunized (1 μg of Spikevax, Moderna) once and had detectable titers of neutralizing antibodies to the index virus SARS-CoV-2$^{D614G}$ 1 week prior to infection compared to the control groups (Fig. 4a; Supplementary Fig. 10a). Naïve and immunized mice were infected intranasally with $10^4$ TCID50/mouse of SARS-CoV-2$^{S-Delta}$, SARS-CoV-2$^{S-Omicron}$, or SARS-CoV-2$^{D614G}$.

Interestingly, only unvaccinated mice inoculated with SARS-CoV-2$^{S-Delta}$, or SARS-CoV-2$^{D614G}$ showed reduction in body weight at 6 dpi, while unvaccinated mice challenged with SARS-CoV-2$^{S-Omicron}$ and all vaccinated mice, regardless of the challenge virus, did not lose body weight (Fig. 4b, Supplementary Fig. 10b). Accordingly, clinical scores were highest for unvaccinated mice infected with SARS-CoV-2$^{S-Delta}$ (Supplementary Fig. 10c). Viral RNA loads in oropharyngeal swabs and several organs were mostly lower for SARS-CoV-2$^{S-Omicron}$-infected mice and remained high for SARS-CoV-2$^{S-Delta}$ in the nasal cavity (Fig. 4c). Nevertheless, viral RNA loads were in most cases reduced at 6 dpi in vaccinated mice, although, as expected, reduction was less pronounced in SARS-CoV-2$^{S-Delta}$- and SARS-CoV-2$^{S-Omicron}$-infected mice (Fig. 4c). Strikingly, the detection of infectious virus in the nose, lung and brain of infected naïve and vaccinated mice perfectly illustrated the effect of vaccination and phenotypic differences between SARS-CoV-2$^{D614G}$, SARS-CoV-2$^{S-Delta}$ and SARS-CoV-2$^{S-Omicron}$ (Fig. 4d). Infectious wild-type SARS-CoV-2$^{D614G}$ is detected in unvaccinated mice in the nose and lung and later at 6 dpi at lower levels also in the brain. However, mRNA immunization efficiently restricted SARS-CoV-2$^{D614G}$ replication, as infectious virus was only detectable at 2 dpi in the nose and not in any other tissue. SARS-CoV-2$^{S-Delta}$ titers in unvaccinated mice were comparable to SARS-CoV-2$^{D614G}$ titers in the nose, while we observed higher titers in the lung and particularly in the brain (Fig. 4d, Supplementary Fig. 10e, f). As expected, SARS-CoV-2$^{S-Delta}$ showed some

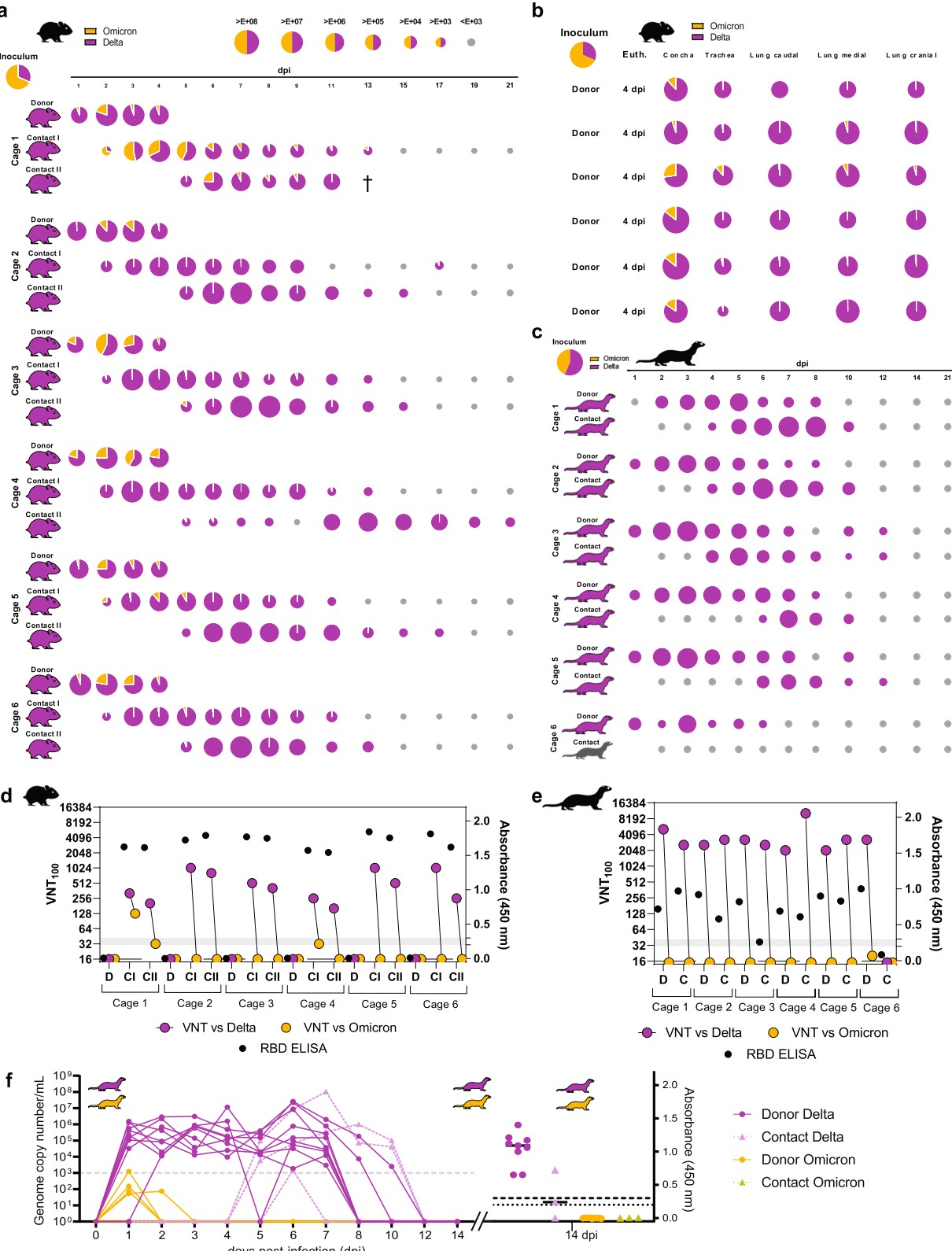

degree of immune escape as infectious virus was readily detectable at 2 dpi in the nose and lung, but eventually was cleared at 6 dpi in vaccinated mice (Fig. 4d). Finally, we detected less infectious virus of SARS-CoV-2^S-Omicron compared to SARS-CoV-2^D614G and SARS-CoV-2^S-Delta in the nose, lung and brain of unvaccinated mice, suggesting that the Omicron-BA.1 spike gene confers a less virulent phenotype than the 614G and the Delta spike gene (Fig. 4d, Supplementary Fig. 10e, f).

Moreover, the Omicron-BA.1 spike gene conferred the largest degree of immune evasion, since infectious titers were comparable between vaccinated and unvaccinated mice in the nose at 2 and 6 dpi and in the lung at 2 dpi. However, no infectious virus was detected in the brains of vaccinated mice, and titers in the lungs were reduced compared to unvaccinated mice at 6 dpi, suggesting that the mRNA vaccine is still of advantage to combat SARS-CoV-2^S-Omicron infection in the LRT and

**Fig. 2 | In vivo competitive co-infection and single infection studies with VOC Delta and Omicron-BA.1 in Syrian hamsters and ferrets.** Simultaneous co-inoculation of six donor hamsters and ferrets each with a Delta:Omicron-BA.1 mixture (hamster, ratio of 1:2.16, total $10^{4.5}$ TCID$_{50}$/hamster and ferrets, ratio of 1:1.43, total $10^{4.75}$ TCID$_{50}$/ferret) and sequential pairwise co-housing of contact animals. All data were quantified using RT-qPCR. Pie chart size represents the total amount of viral RNA (vRNA) detected in each sample (exact vRNA equivalents are found in Source Data files and the coloring shows the individual VOC ratio). Animal silhouettes are colored according to the dominant (>66%) VOC. Limit of detection was set at $10^3$ vRNA copies per mL. **a** Nasal washings of Donor and Contact I + II Syrian hamster pairs from 1 to 21 dpi. **b** vRNA in URT and LRT of donor hamsters at 4 dpi. **c** Nasal washings of Delta and Omicron-BA.1 co-inoculated Donor and respective Contact ferrets for the 21-day infection period. **d, e** Antibody detection in hamsters (**d**) and ferrets (**e**) via VNT$_{100}$ and RBD-ELISA after simultaneous Delta and Omicron-BA.1 co-inoculation shown for donor (D), contact (C), contact I (CI), and contact II (CII) animals. Specific neutralizing capacity of sera against the Delta (pink) and Omicron-BA.1 (yellow) virus pair were analyzed. Reactivity of sera below 1:32 pre-dilution was considered negative. Generalized seroreaction was also determined by RBD-ELISA (black dots). **f** vRNA detection and seroreactivity in ferrets after infection with single virus (Delta or Omicron-BA.1); donor animals (solid line) ($n = 9$) and contact animals (dashed line) ($n = 3$). Limit of detection was set at $10^3$ vRNA copies per mL and for antibody detection at >0.2 (questionable) and >0.3 (positive). Source data are provided as a Source Data file 2.Source Data file 2.

systemic dissemination (Fig. 4d, Supplementary Fig. 10e, f). The different virus phenotypes and vaccine efficiencies are corroborated by the pathological findings (Fig. 4e, Supplementary Fig. 10d; Supplementary Information Table 3). Unvaccinated mice infected with SARS-CoV-2$^{S-Delta}$ displayed severe interstitial lymphohistiocytic pneumonia with concurrent vascular inflammation and widespread nucleocapsid antigen detection in the immunohistochemical (IHC) analysis starting already at 2 dpi, while infection with SARS-CoV-2$^{D614G}$ resulted in similarly severe pathological findings only at 6 dpi. In contrast, unvaccinated mice that were infected with SARS-CoV-2$^{S-Omicron}$ displayed milder histopathological lung lesions and less nucleocapsid antigen was detected by IHC analysis. In agreement with the observed reduced virus titers in the lungs of vaccinated mice, we observed milder lung histopathological lesions and almost no nucleocapsid antigen IHC detection in vaccinated mice with any of the viruses when compared to lungs of unvaccinated mice.

Collectively, these findings demonstrate the major impact of the Delta and Omicron-BA.1 spike genes on virus replication, immune evasion, and pathogenicity. Compared to the progenitor D614G spike gene, the Delta spike gene confers increased replication, pathogenicity, and immune escape. The Omicron-BA.1 spike gene is conferring the greatest degree of immune evasion, compared to the wild-type D614G spike and the Delta spike genes, resulting in comparable or even increased detection of infectious SARS-CoV-2$^{S-Omicron}$ compared to SARS-CoV-2$^{D614G}$ and SARS-CoV-2$^{S-Delta}$, in several tissues and organs of vaccinated mice. Importantly, the Omicron-BA.1 spike gene also confers reduced pathogenicity, as seen in unvaccinated mice (Fig. 4b, e), suggesting that the Omicron-BA.1 spike is a major determinant of the observed milder disease in humans.

## Discussion

The appearance of Omicron-BA.1 in the human population exemplified a remarkable jump in SARS-CoV-2 evolution. Omicron-BA.1 has acquired up to 50 mutations of which at least 34 are located within the spike gene[3]. Particularly, the NTD and RBD harbor many mutations that have not been seen in previous SARS-CoV-2 variants, suggesting substantial changes in spike antigenicity, receptor binding, and possibly other spike functions. Accordingly, the current experimental systems to assess phenotypic changes of SARS-CoV-2 require a revision regarding the extent to which they truly reflect epidemiological and clinical observations in humans. We and others previously showed that the Syrian hamster is a highly susceptible animal model for several SARS-CoV-2 variants and showed a great efficiency in replication and transmission[9,10,13]. Nevertheless, we show here that competitive infection and transmission experiments in the hamster model no longer reflect the human epidemiological situation. While in humans, Omicron-BA.1 rapidly became the prevailing variant over Delta, the variant that previously has outcompeted Alpha, we observed that in competitive infection experiments in the hamster model the exact opposite order of variants is dominant. Certainly, these differential observations hint to an adaptation of SARS-CoV-2

variants to the human host. Our data show that Alpha is massively replicating and dominantly transmitted with severe clinical features in the hamster model. Moreover, replicative fitness of Delta and Omicron-BA.1 decreased sequentially in direct competition with the respective earlier predominant variant in the human population[14,15]. These observations are in line with the high degree of mortality (83 % mortality) in the Delta vs Alpha compared to the Delta vs Omicron-BA.1 competition experiment (8 % mortality). A comparable high mortality rate was also seen in association with Alpha in our earlier study of paired VOC competitions including Alpha, Beta, and SARS-CoV-2$^{D614G}$ [10], suggesting that the degree of pathogenicity in the Syrian hamster correlates with the respective replicative fitness in this model.

Another surprising finding in our study was that Omicron-BA.1 does not productively replicate in ferrets. While we see, in agreement with the epidemiological situation in humans, that the Delta variant vastly predominates over the Alpha variant in terms of early replication and especially transmission in the ferret model, we observed that Omicron-BA.1 infection is abortive. The remarkable observations of Omicron-BA.1 being outcompeted by the Delta variant in naïve Syrian hamsters and the complete block of ferret susceptibility towards Omicron-BA.1 provides further evidence to an adaptation of this variant towards humans. This also makes it rather unlikely that the Omicron variant evolved in an animal reservoir.

It was therefore important to include animal models that are based on the usage of the authentic hACE2 receptor. The hACE2 knock-in mice are a valuable model for SARS-CoV-2 replication in the URT as they show robust SARS-CoV-2 replication with only mild or sub-clinical disease[10,11]. The observation that also in this model Delta outcompeted Omicron-BA.1 and similarly, the corresponding Delta spike clone SARS-CoV-2$^{S-Delta}$ outcompeted the Omicron-BA.1 spike clone SARS-CoV-2$^{S-Omicron}$, suggests that the intrinsic replicative fitness of Delta is still higher than that of Omicron-BA.1 once a productive replication is established within the infected host and that this phenotype is mediated by the spike genes of Delta and Omicron-BA.1. Notably, the infection of hACE2-KI mice with the Delta isolate resulted for the first time in our hands in evident lung pathology and body weight loss, suggesting that this model recapitulates the increased pathogenicity of Delta in humans and that it will receive further attention in future studies that aim at assessing SARS-CoV-2 variants with high pathogenicity.

Notably, by using hNEC and hBEC cultures we were able to detect an accelerated growth of Omicron-BA.1 in hNECs within the first 48 h post infection that reflects the observed shorter incubation period of Omicron-BA.1 versus previous SARS-CoV-2 variants in humans. In contrast, in hBECs, Omicron-BA.1 replication is limited to lower peak titers, suggesting reduced replication in bronchial and lung tissue as previously demonstrated in PCLS ex vivo tissue cultures[16]. Importantly, this phenotype is also seen by infections with the Omicron-BA.1 spike clone SARS-CoV-2$^{S-Omicron}$, demonstrating that it is spike-mediated. Moreover, the reduced replication of

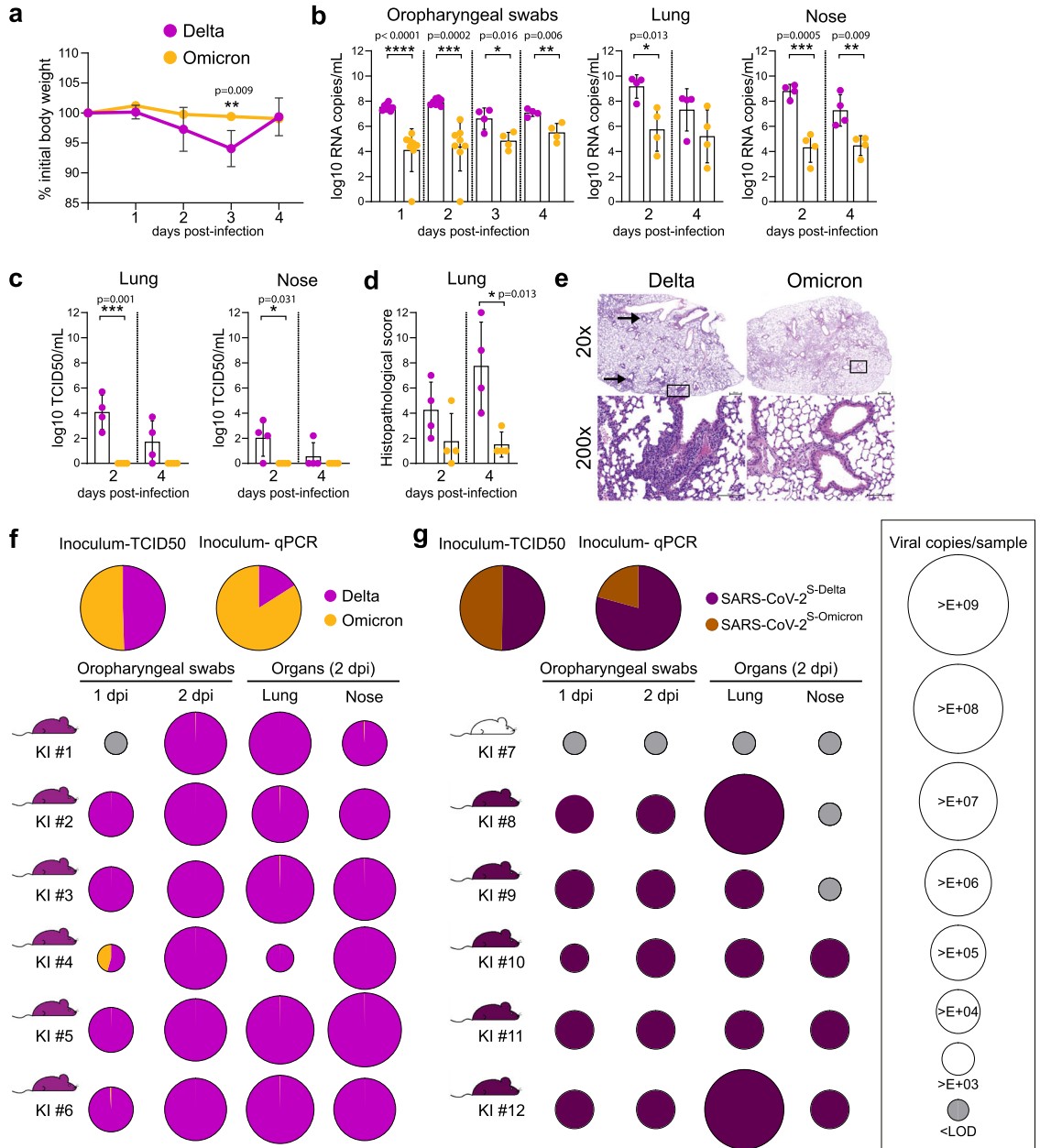

**Fig. 3 | Delta spike mutations drive enhanced fitness in hACE2-knock-in mice.**
**a**–**e** hACE2-KI mice (7–16 week-old male, $n = 8$ mice/virus) were intranasally inoculated with $10^{4.3}$ TCID$_{50}$ of Delta or Omicron. **a** Average relative weight loss after infection (Relative weight loss data from each animal is given in Supplementary Fig. 9). **b** Viral copies per mL of oropharyngeal swabs or per lung and nose sample ($n = 16$ mice) quantified using E-gene probe-specific RT–qPCR. **c** Infectious virus titers from the lung and nose samples ($n = 16$ mice) determined using TCID$_{50}$ assays in VeroE6/TMPRSS2 cells. **d**, **e** Histopathological score and hematoxylin and eosin staining from Delta- and Omicron-infected lung sections ($n = 16$ mice) at 2 and 4 dpi. Perivascular and peribronchiolar lymphohistiocytic inflammation are highlighted with an arrow, and the higher magnification represented in the lower panel corresponds to the areas highlighted by a square in the upper panel. Scale bars, 500 (upper panel) and 100 μm (lower panel). Data are mean ± s.d. from the indicated number of biological replicates from a single experiment. The color key in a also applies to b, c and d. Statistical significance was determined using an unpaired two-

tailed Student $t$-test; *$P < 0.05$, **$P < 0.01$, ***$P < 0.001$, ****$P < 0.0001$. **f**, **g** hACE2-KI mice (7–19 week-old female, $n = 6$ mice/group) were intranasally inoculated with $10^{4}$ TCID$_{50}$ of a 1:1 mix of **f** Delta and Omicron, or **g** SARS-CoV-2$^{S\text{-Delta}}$ and SARS-CoV-2$^{S\text{-Omicron}}$. qPCR quantification of the ratio of the two variants or recombinant viruses present in the inoculum is reported. Oropharyngeal swabs were collected 1 and 2 dpi; lung and nose tissues were collected on 2 dpi. Pie charts show the ratio of variants detected in each sample at the indicated dpi ($n = 6$ mice/group). Pie chart sizes are proportional to the total number of viral genome copies per ml (swabs) or per sample (tissues), as shown in the legend on the right. Gray pies indicate values below the LOD (i.e., $10^{3}$ viral RNA copies per mL/sample). Mouse silhouettes are colored to indicate the dominant SARS-CoV-2 variant (>66%) in the last positive swab sample from the corresponding mouse. KI numbers 1–12 denote individual hACE2-KI mice. Data was obtained from one experiment. Source data are provided as a Source Data file 3.Source Data file 3.

SARS-CoV-2$^{S\text{-Omicron}}$ in the lungs of K18-hACE2 mice and the resulting mild pathology confirms that this phenotype is indeed spike-mediated and likely contributes to the reduced pathogenicity of Omicron-BA.1 as seen in humans. Of note, the increased replication, pathology, and clinical scores of SARS-CoV-2$^{S\text{-Delta}}$ point to the

opposite phenotype of increased pathogenicity that is according to our study also mediated by the spike gene.

Finally, our study illustrates the degree of immune evasion of Omicron-BA.1 in comparison to Delta and the index virus SARS-CoV-2$^{D614G}$. The mRNA vaccine is well matched to the index virus SARS-CoV-

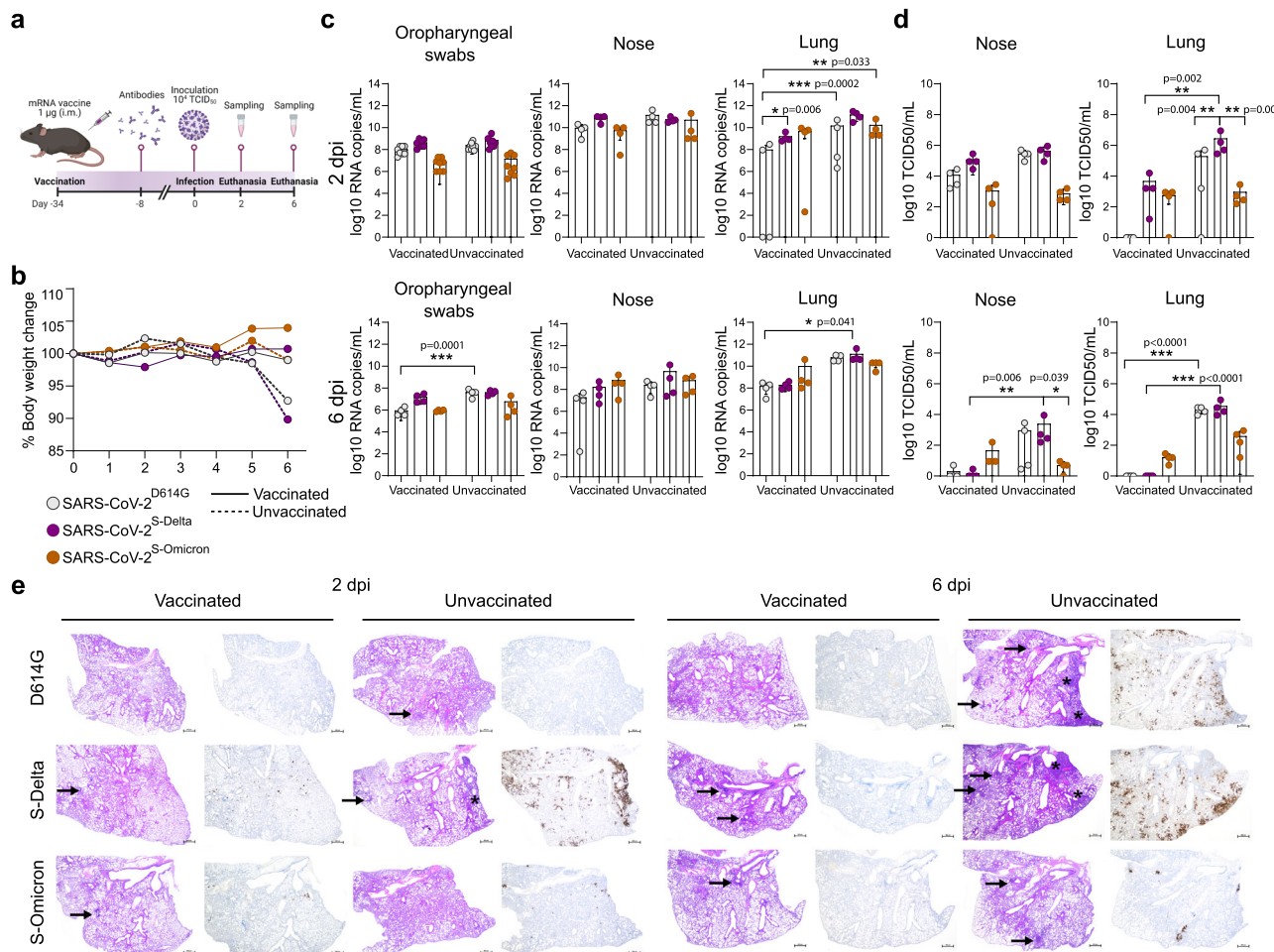

**Fig. 4 | mRNA vaccine induced reduction in replication and pathogenesis of SARS-CoV-2 clones in K18-hACE2 transgenic mice. a** Female K18-hACE2 transgenic mice (7–15 weeks old, $n = 8$ mice/group) were immunized intramuscularly with a single dose of 1 µg of mRNA-Vaccine Spikevax (Moderna). After 2 weeks the neutralizing antibody titers against SARS-CoV-2 were determined (Supplementary Fig. 10a). Later, mice were intranasally inoculated with $10^4$ TCID$_{50}$ of SARS-CoV-2$^{D614G}$, SARS-CoV-2$^{S-Delta}$, or SARS-CoV-2$^{S-Omicron}$. Body weight change and clinical scores of the mice were monitored daily. **b** The mean body weight change is presented (Data from individual animals are shown in Supplementary Fig. 10b). Only the unvaccinated mice infected with SARS-CoV-2$^{D614G}$ and SARS-CoV-2$^{S-Delta}$ showed noticeable weight loss. **c** Oropharyngeal swabs, lung and nose samples of the infected mice were collected at 2 or 6 days post-infection (dpi) to determine the viral load ($n = 4$ for each virus). Viral RNA-dependent RNA polymerase (RdRp) gene copies were quantified using probe-specific RT–qPCR. **d** Infectious virus titers from

the lung and nose samples ($n = 8$ mice/group) were determined using TCID$_{50}$ assays in VeroE6/TMPRSS2 cells. **e** Hematoxylin and eosin stain (left panel) and immunohistochemical analysis specific for SARS-CoV-2 nucleocapsid protein (right panel) of lung sections in vaccinated (A) and unvaccinated mice (B) at 2 and 6 dpi following infection with SARS-CoV-2$^{D614G}$ ($n = 3$), SARS-CoV-2$^{S-Delta}$ ($n = 3$), and SARS-CoV-2$^{S-Omicron}$ ($n = 4$). Consolidated lung areas are highlighted with a star, and perivascular and peribronchiolar lymphohistiocytic inflammation highlighted with an arrow. Scale bars, 500 µm. Data are mean ± s.d. from the indicated number of biological replicates. The color key in b also applies to c and d. Statistical significance was determined using two-way ANOVA (**a**–**d**) and $P$-values were adjusted using Tukey's multiple-comparison test; *$P < 0.05$, **$P < 0.01$, ***$P < 0.001$, ****$P < 0.0001$. Data were obtained from one experiment. Each data point represents one biological replicate. Source data are provided as a Source Data file 4.Source Data file 4.

$2^{D614G}$, and accordingly, vaccinated mice are well protected. Recently, it has been shown that the replicative and transmissive fitness advantage of Omicron against Delta changes in favor of Omicron when hamsters are vaccinated, highlighting the influence of Omicron-associated immune escape potential and the importance of the immune status on virus selection[17]. Our data from vaccinated mice extend this finding by using an animal model with the authentic hACE2 receptor, and by assigning this context-specific phenotypic change to the Omicron-BA.1 spike gene.

In summary, we provide here a comprehensive and comparative analysis of the Omicron-BA.1 phenotype by using several advanced in vitro and in vivo systems and different VOCs. We demonstrate that Omicron-BA.1 displays a remarkable evolutionary and phenotypic jump that impacts virus replication, host and tissue tropism, pathogenicity, and immune escape, with the spike gene being a key determinant of these phenotypic changes.

## Methods
### Biosafety statement
All experiments with infectious SARS-CoV-2 were performed in enhanced biosafety level 3 (BSL3) containment laboratories at Institute of Virology and Immunology, Mittelhäusern, Switzerland, and Friedrich-Loeffler-Institut, Greifswald-Insel Riems, Germany, which followed the approved standard operating procedures of BSL3 facility of relevant authorities in Switzerland and Germany. Before commencing work, all personnel received relevant training.

### Cells and culture conditions
At IVI, IBSC VeroE6 (Vero C1008, ATCC) and VeroE6/TMPRSS2 cells (NIBSC Research Reagent Depository, UK) were cultured in Dulbecco's modified Eagle's medium (DMEM). BHK-SARS-N (BHK-21 cells expressing the N protein of SARS)[18] were grown in minimal essential medium (MEM). Both media were supplemented with 10% (v/v) fetal bovine

serum, 1% (w/v) non-essential amino acids, 100 IU/mL penicillin, 100 µg/mL streptomycin µg/ml and the cell lines maintained at 37 °C in a 5% CO$_2$ atmosphere.

VeroE6 cells at FLI (Collection of Cell Lines in Veterinary Medicine CCLV-RIE 0929) were cultured using a mixture of equal volumes of Eagle MEM (Hanks' balanced salts solution) and Eagle MEM (Earle's balanced salts solution) supplemented with 2 mM L-Glutamine, non-essential amino acids adjusted to 850 mg/L, NaHCO$_3$, 120 mg/L sodium pyruvate, 10% fetal bovine serum (FBS), pH 7.2.

Calu-3 cells (HTB-55, American Type Culture Collection (ATCC), Manassas, VA, USA) were propagated in Dulbecco's modified Eagle Medium–GlutaMAX, supplemented with 10% (v/v) heat-inactivated fetal bovine serum, 100 mg/mL streptomycin, 100 IU/mL penicillin, 1% (w/v) non-essential amino acids, and 15 mM 4-(2-hydroxyethyl)–1-piperazineethanesulfonic acid (HEPES, Gibco, Gaithersburg, MD, USA). Cells were maintained at 37 °C in a humidified incubator with 5% CO$_2$.

## Viruses
Viruses are listed in Supplementary Table 1. Viruses were cultivated on VeroE6, VeroE6/TMPRSS2, or Calu-3 cells and sequence verified by performing whole-genome NGS sequencing (see below). For the hamster and ferret infection studies SARS-CoV-2 Alpha (hCoV-19/Germany/NW-RKI-I-0026/2020, L4549, SARS-CoV-2 B.1.1.7 NW-RKI-I-0026/2020 passage 3 of EPI_ISL_751799), one silent mutation in the ORF1a (sequence position 11741), SARS-CoV-2 Delta AY.127 (hCoV-19/Switzerland/BE-IFIK-918-4879/2021, L5109, passage of EPI_ISL_1760647) and SARS-CoV-2 Omicron-BA.1 (BA.1 (hCoV-19/Germany/HE-FFM-30318738/2021, passage of EPI_ISL_6959868)) were used. The Omicron-BA.1 isolate was from the Institute of Medical Virology, University Hospital Frankfurt, Goethe University, Frankfurt am Main, Germany. For the hamster and ferret competition experiments, respective Alpha, Delta or Omicron-BA.1 viruses were propagated (three passages for Alpha, two passages for Omicron-BA.1, one passage for Delta) on VeroE6 cells (Collection of Cell Lines in Veterinary Medicine CCLV-RIE 0929) using a mixture of equal volumes of Eagle MEM (Hanks' balanced salts solution) and Eagle MEM (Earle's balanced salts solution) supplemented with 2 mM L-Glutamine, nonessential amino acids adjusted to 850 mg/L, NaHCO3, 120 mg/L sodium pyruvate, 10% fetal bovine serum (FBS), pH 7.2. The virus was harvested after 72 h, titrated on VeroE6 cells and stored at −80 °C until further use.

For in vitro experiments, Delta and Omicron-BA.1 were isolated at the University of Bern. Briefly, an aliquot of 250 µl of SARS-CoV-2 nasopharyngeal swab samples from positive patients (Delta; EPI_ISL_1760647, Omicron-BA.1; EPI_ISL_7062525) were centrifuged for 5 min at room temperature at 200 × g. 200 µl of clinical material was transferred to confluent Calu-3 cells and incubated for 2–3 days at 37 °C in a humidified CO$_2$-incubator (5%). Virus containing supernatant was cleared from cell debris through centrifugation for 5 min at 500 × g before aliquoting and storage at −80 °C. All virus stocks were sequenced with Nanopore sequencing technology using a revised ARTIC midnight protocol (Fragment 28 update) allowing sequencing of both Delta and Omicron-BA.1 variants. Sequence verified stocks at passage 3 were used. For the experiments in hACE2-KI mice, the Delta isolate (EPI_ISL_2535433)[19] was kindly provided by Georg Kochs, Institute of Virology, Freiburg, Germany and Omicron-BA.1 was isolated at the University of Bern (EPI_ISL_7062525). The TCID$_{50}$ titers have been determined on VeroE6 and were calculated according to the Spearman-Kaerber formula.

## Generation of infectious cDNA clones using transformation-associated recombination cloning and rescue of recombinant viruses
The generation of recombinant SARS-CoV-2 was approved by the Swiss Federal Office for the Environment (A202819-01). We used the in-yeast transformation-associated recombination (TAR) cloning method as

described previously with a few adaptations to generate SARS-CoV-2$^{S-Delta}$ and SARS-CoV-2$^{S-Omicron}$ [20]. Briefly, the whole SARS-CoV-2 genome was encoded in 12 overlapping DNA fragments. These so-called WU-Fragments and a TAR-vector are homologously recombined in yeast forming the yeast artificial chromosome (YAC). WU-Fragments 9 and 10 covering the spike region were replaced by newly generated and overlapping PCR products. To introduce the variant specific mutations into the spike gene, we used 50 bp primers containing the desired nucleotide changes in combination with YAC DNA templates from previously cloned viruses (Supplementary Information Table 4). Also, by using these 50 bp long primers homologous overlaps between the PCR products were created. Six PCR reactions using the Q5® High-Fidelity DNA Polymerase (NEB) were performed to replace WU-Fragment 9 and 10 to create the SARS-CoV-2$^{S-Delta}$. To create the SARS-CoV-2$^{S-Omicron}$ and its sub spike clones overlapping PCR products via RT from Omicron RNA template were done. In brief, cDNA was generated from RNA (Omicron-BA.1; EPI_ISL_7062525) by LunaScript RT SuperMix (NEB). PCR reactions using Q5® High-Fidelity DNA Polymerase were performed with the primers and templates described in Supplementary Information Table 4. The resulting PCR products were mixed and matched for Omicron-Spike, -NTD, -RBD and -CS to replace WU-Fragment 9 and 10. All PCR products were purified by the High Pure PCR Product Purification Kit (Roche) before being used for TAR cloning.

In vitro transcription was performed for EagI-cleaved YACs and PCR-amplified SARS-CoV-2 N gene using the T7 RiboMAX Large Scale RNA production system (Promega) as described previously[20]. Transcribed capped mRNA was electroporated into baby hamster kidney (BHK-21) cells expressing SARS-CoV N protein. Electroporated cells were co-cultured with susceptible VeroE6/TMPRSS2 cells to produce passage 0 (P.0) of the recombinant viruses. Subsequently, progeny viruses were used to infect fresh VeroE6/TMPRRS2 cells to generate P.1 stocks for downstream experiments.

## Virus replication kinetics on human primary airway cells
hNEC and hBEC cultures were infected with 10$^4$ TCID$_{50}$ of the SARS-CoV-2 variants listed in the Supplementary Table 1. Viruses were diluted in HBSS (Gibco), inoculated on the apical side, and incubated for 1 h at 33 °C in case of hNECs or 37 °C in case of hBECs. Subsequently, the inoculum was removed, and the cells were washed three times with 100 µl of HBSS. The third wash was collected as the 1 hpi time point. For the duration of the experiment, hNECs and hBECs were incubated in a humidified incubator with 5% CO$_2$ at 33 °C or 37 °C, respectively. To measure virus progeny release, apical washes were performed every 24 h up to 96 hpi. 100 µl HBSS were incubated on the apical side for 10 min prior to the respective time point and subsequently collected, diluted 1:1 with virus transport medium (VTM), and stored at −80 °C for later analysis.

Virus titers were assessed by standard TCID$_{50}$ assays on VeroE6/TMPRSS2 cells. In short, 2 × 10$^4$ cells/well were seeded in a 96-well plate 1 day before the titration and were then inoculated with a 10-fold serial dilution of the prior collected apical washes. Four technical replicates were performed for each sample. Cells were then incubated for 72 h at 37 °C in a humidified incubator with 5% CO$_2$. Subsequently, cells were fixed with 4% (v/v) buffered formalin (formafix) and stained with crystal violet. Infected wells were counted manually, registered in Microsoft Excel 2016 (16.0.5239.1001), and TCID$_{50}$ was calculated according to the Spearman-Kaerber formula.

## Competition assay in hNEC and hBEC cultures
Inoculum mixtures were generated by mixing the respective viruses at a 1:1 ratio based on genome equivalents (GE) determined by qPCR including RNA standard. Each mixture contained 6 × 10$^7$ GE of each respective virus. hNECs and hBECs were infected with inoculum mixtures apically and incubated for 1 h at 33 °C or 37 °C respectively.

Afterwards, inocula were removed and the cells were washed three times with 100 µl HBSS (Gibco). For the duration of the experiment, the hNECs and hBECs were incubated in a humidified incubator with 5% $CO_2$ at 33 °C or 37 °C, respectively. Apical washes were performed and collected at 2, 4 and 6 dpi. 100 µl HBSS were incubated on the apical side for 10 min prior to the respective time point and subsequently collected, mixed with 300 µl DNA/RNA Shield lysis buffer (Zymo Research) and stored at −80 °C for later analysis.

## Nanopore sequencing workflow
Virus stocks, inoculum mixtures, and samples from competition assays in NECs, BECs, and lung slices were sequenced using the MinION sequencer (Oxford Nanopore Technologies) following the ARTIC nCoV-2019 sequencing protocol V3 (LoCost) (https://protocols.io/view/ncov-2019-sequencing-protocol-v3-locost-bh42j8ye) with the following modifications: the Midnight primer scheme (1200 bp amplicons) was used to perform the multiplex PCR (https://www.protocols.io/view/sars-cov2-genome-sequencing-protocol-1200bp-amplic-rm7vz8q64vx1/v6) instead of the ARTIC V3 primer scheme. In addition, two extra Omicron-specific primers (SARSCoV_1200_Omicron_24_L: 5′-GCT GAA TAT GTC AAC AAC TCA TAT GA-3′ and SARSCoV_1200_Omicron_28_L: 5′-TTT GTG CTT TTT AGC CTT CT GTT-3′) were added to Pool 2 of the multiplex PCR to achieve similar levels of amplification for all viruses sequenced.

RNA was extracted for all samples using either the Quick-RNA Viral Kit (Zymo Research) or the NucleoMag VET kit (Machery-Nagel) according to the manufacturer's guidelines on a Kingfisher Flex Purification system (Thermo fisher). Extracted RNA was assessed using the TaqPath COVID-19 CE-IVD RT-PCR Kit (Thermo fisher) and cDNA was prepared using the LunaScript RT SuperMix Kit (Bioconcept). Subsequently, a multiplex PCR was used to generate overlapping 1200 bp amplicons that span the length of the SARS-CoV-2 genome for all used virus VOCs. The Q5 Hot Start High-Fidelity 2X Master Mix (Bioconcept) was used for the multiplex PCR reaction. For library preparation, all samples were barcoded using the Native Barcoding Kit 96 (Oxford Nanopore Technologies, SQK-NBD112-96). Libraries were then loaded onto a R9.4.1 flow cell on a MinION sequencer (Oxford Nanopore Technologies) and monitored using the MinKNOW software (Version 21.11.9). A no-template negative control from the PCR amplification step was prepared in parallel and sequenced on each flow cell.

Live GPU basecalling was performed using Guppy v.5.1.15 (Oxford Nanopore technologies) in high-accuracy mode. Following sequencing, downstream analysis was performed using a modified version of the nCoV-2019 novel coronavirus bioinformatics protocol (ARTIC Network, https://artic.network/ncov-2019/ncov2019-bioinformatics-sop.html). The command 'artic gupplex' was used to filter "pass" reads based on length with–max-length set to 1400. The 'artic minion' command was then used to align the filtered reads to the Wuhan-Hu-1 reference genome (accession MN908947.3 [www.ncbi.nlm.nih.gov/bioproject/?term=MN908947.3]) with the 'normalize' parameter set to 500. BAM alignment files generated using the ARTIC pipeline were subsequently used as input to call variants in longshot (v.0.4.4). An input VCF file containing VOC Delta and Omicron BA.1 mutations was provided to longshot in order to genotype specific nucleotide sites. Output VCF files for each sample were used as input for downstream analysis in R v.4.1.3. Calculations were performed on UBELIX (http://www.id.unibe.ch/hpc), the HPC cluster at the University of Bern.

Downstream analysis of VCF files for each sample was performed using a custom script in R v.4.1.3. Briefly, each VCF file was first filtered to exclude mutations shared between both viruses in the mixture, non-lineage-defining mutations, sites that were difficult to call (e.g., sites with unique but overlapping mutations or with a high number of ambiguous calls), and mutations called with a depth of coverage <100. Sequencing depth across the entire genome was also checked for each

sample along with the frequency of any 'shared' mutations in each virus mixture (in these cases the frequency should be close to 1). Following filtering and quality control checks, the remaining variant calls were used to calculate the mean mutation frequencies for each virus on a per sample basis (Supplementary Fig. 1). Finally, the mean ± sd virus ratio was calculated for each virus mixture at each time point for NEC, BEC, and lung slices.

## Ion torrent sequencing
Virus stocks was sequenced using a generic metagenomics sequencing workflow as described previously[21] with some modifications. For reverse-transcribing RNA into cDNA, SuperScriptIV First-Strand cDNA Synthesis System (Invitrogen, Germany) and the NEBNext Ultra II Non-Directional RNA Second Strand Synthesis Module (New England Biolabs, Germany) were used, and library quantification was done with the QIAseq Library Quant Assay Kit (Qiagen, Germany). Libraries were sequenced using an Ion 530 chip and chemistry for 400 base pair reads on an Ion Torrent S5XL instrument (Thermo Fisher Scientific, Germany).

## Ethics statements for human subjects and animal experimentation
All ferret and hamster experiments were evaluated by the responsible ethics committee of the State Office of Agriculture, Food Safety, and Fishery in Mecklenburg–Western Pomerania (LALLF M-V) and gained governmental approval under registration number LVL MV TSD/7221.3-1-004/21.

Mouse studies were approved by the Commission for Animal Experimentation of the Cantonal Veterinary Office of Bern and conducted in compliance with the Swiss Animal Welfare legislation and under license BE43/20.

Lung tissue for the generation of human precision-cut lung slices (PCLSs) and human bronchial epithelial cells (hBECs) was obtained from patients undergoing pulmonary resection at the University Hospital of Bern, Inselspital, Switzerland, and the Cantonal Hospital of St. Gallen, Switzerland, respectively. Written informed consent was obtained for all the patients and the study protocols were approved by the respective local Ethics Commissions (KEK-BE_2018-01801, EKSG 11/044, and EKSG 11/103).

## Hamster competition studies
The study outline for the hamster competition studies can be seen in Supplementary Fig. 2a. Six Syrian hamsters (*Mesocricetus auratus*) (Janvier Labs) were inoculated intranasally under a brief inhalation anesthesia with a 70 µl mixture of two respective SARS-CoV-2 VOC (Alpha versus Delta and Delta versus Omicron-BA.1), referred to as donor hamsters. Each inoculum, as well as the single viruses were backtitrated followed by determination of VOC ratio by dividing the $TCID_{50}$/mL values of single VOC1 by VOC2. One day following inoculation of the donor hamsters, we co-housed six naïve contact hamsters (Contact I) in a 1:1 setup, allowing direct contact of donor to contact hamster. The donor hamsters were removed from the experiment on 4 dpi for organ sampling (RT-qPCR) and have been replaced by another naïve contact hamster (Contact II).

Viral shedding was monitored by nasal washings in addition to a daily physical examination and body weighing routine. Nasal washing samples were obtained under a short-term isoflurane inhalation anesthesia from individual hamsters by administering 200 µl PBS to each nostril and collecting the reflux. Animals were sampled daily from 1 dpi to 9 dpi and afterwards every second day until 21 dpi. Under euthanasia, serum samples and an organ panel comprising representative upper (URT) and lower respiratory tract (LRT) tissues were collected from each hamster. All animals were checked daily for signs of clinical disease and weight loss. Animals reaching the humane endpoint, e.g., falling below 80% of the initial body weight relative to 0 dpi, were humanely euthanized.

## Ferret competition studies

The study outline for the ferret competition studies can be seen in Supplementary Fig. 2b. Similar to the hamster study, 12 ferrets (six donor ferrets and six contact ferrets) from the FLI in-house breeding were housed pairwise in strictly separated cages to prevent spillover contamination. Of these, six ferrets were intranasally inoculated with an equal 250 µl mixture of SARS-CoV-2 Alpha and Delta or Delta and Omicron-BA.1. The inoculum of the mixture as well as from the single viruses was back titrated and the ratio of each variant was determined by dividing the $TCID_{50}$/mL values of single VOC1 by VOC2. Ferret pairs were separated for the first 24 h following inoculation. Subsequently, the ferrets were co-housed again, allowing direct contact of donor to contact ferrets. All ferrets were sampled via nasal washings with 750 µl PBS per nostril under a short-term isoflurane inhalation anesthesia. All ferrets, which were in the study group on the respective days, were sampled daily until 8 dpi and afterwards every second day until the animals were negative for SARS-CoV-2 viral genome in RT-qPCR and one last time at the study end (21 dpi). Physical condition of all animals was monitored daily throughout the experiment.

## Ferret single infection studies

The study outline for the ferret single infection studies can be seen in Supplementary Fig. 2c. 12 ferrets (nine donor and three contact animals) from the FLI in-house breeding were housed in multiple connected cage units. The donor ferrets were inoculated either with 250 µl of SARS-CoV-2 Delta ($10^{4.8125}$ $TCID_{50}$/ferret, calculated from back-titration of the original material) or Omicron-BA.1 ($10^{5.125}$ $TCID_{50}$/ferret, calculated from back-titration of the original material) in two separate and independent animal trials. Contact animals were separated from the donor animals for the first 24 h, followed by co-housing again to allow direct contact of donor and contact animals. All ferrets were sampled via nasal washings with 750 µl PBS per nostril under a short-term isoflurane inhalation anesthesia. Sampling was done daily until 8 dpi and afterwards every second day until the study end at 14 dpi. For serological analysis, serum was collected at the study end (14 dpi). Physical condition of all animals was monitored daily throughout the experiment. For analysis of SARS-CoV-2 Omicron-BA.1 viral genome distribution in LRT and URT, six ferrets from the respective Omicron-BA.1-single infection trial were euthanized at 6 dpi and viral organ load was determined via Omicron-BA.1-specific RT-qPCR with Quant-Studio™ Real-Time PCR Software (v1.7.1).

## Mouse studies

Mice were produced at the specific-pathogen-free facility of the Institute of Virology and Immunology (Mittelhäusern), where they were maintained in individually ventilated cages (blue line, Tecniplast), with 12-h/12-h light/dark cycle, $22 \pm 1 °C$ ambient temperature and $50 \pm 5\%$ humidity, autoclaved food and acidified water. At least 7 days before infection, mice were placed in individually HEPA-filtered cages (IsoCage N, Tecniplast).

hACE2-KI mice (B6.Cg-*Ace2*$^{tm1(ACE2)Dwnt}$) and hACE2-K18Tg mice (Tg(K18-hACE2)2Prlmn) were described previously[9,22]. All mice were bred at the specific pathogen-free facility of the Institute of Virology and Immunology and housed as previously described[10]. Mice were anesthetized with isoflurane and inoculated intranasally with 20 µl per nostril. For single-infection experiments, 7–17-week-old male mice were inoculated with a dose of $2 \times 10^4$ $TCD_{50}$/mouse of either Delta (EPI_ISL_2535433) or Omicron-BA.1 (EPI_ISL_7062525) isolates. For competition experiments, 7–19-week-old female mice were inoculated with a mixture inoculum containing the Delta and Omicron-BA.1 isolates or a mixture of the recombinant spike clones SARS-CoV-2$^{S-Omicron}$ and SARS-CoV-2$^{S-Delta}$. Inoculum mixtures were generated by mixing the respective viruses aiming at a 1:1 ratio based $TCID_{50}$/mL titers of the single virus. The ratio of each variant in the prepared inocula was

further determined by standard RT-qPCR. At 2 or 4 dpi, mice were euthanized and organs were aseptically dissected. Systematic tissue sampling was performed as described previously[9].

K18-hACE2 mice (all female, 7–15 weeks old) were immunized intramuscularly with a single dose of 1 µg of mRNA-Vaccine Spikevax (Moderna). Five weeks after immunization, the immunized mice and a group of sex- and age-matched naïve animals were challenged intranasally with 20 µl per nostril with the virus inoculum described in the results section. Euthanasia and organ collection was performed 2 or 6 dpi as described above. All mice were monitored daily for body weight loss and clinical signs. Oropharyngeal swabs were collected daily as described before[10].

## Animal specimens work up, viral RNA detection and quantification

Organ samples of about 0,1 cm³ size from ferrets and hamsters were homogenized in a 1 mL mixture composed of equal volumes of Hank's balanced salts MEM and Earle's balanced salts MEM (containing 2 mM L-glutamine, 850 mg l−1 NaHCO3, 120 mg l−1 sodium pyruvate, and 1% penicillin–streptomycin) at 300 Hz for 2 min using a Tissuelyser II (Qiagen) and were then centrifuged to clarify the supernatant. Organ samples from mice were either homogenized in 0.5 mL of RA1 lysis buffer supplemented with 1% β-mercaptoethanol as described[10].

Nucleic acid was extracted from 100 µl of the nasal washes after a short centrifugation step or 100 µl of organ sample supernatant using the NucleoMag Vet kit (Macherey Nagel). Nasal washings, oropharyngeal swabs and organ samples were tested by virus-variant specific RT-qPCR to analyze the genomic ratio of the two different viruses used for inoculation.

Three specific RT-qPCR assays for SARS-CoV-2 Alpha, Delta and Omicron-BA.1 were designed based on the specific genome deletions within the ORF1 and S gene (Supplementary Information Table 5). Here, virus specific primers were used to achieve a high analytical sensitivity (<10 genome copies/µl template) of the according PCR assays, also in samples with a high genome load of the non-matching virus. For each specific RT-qPCR a dilution row of a standard with known concentration determined by digital droplet PCR was carried along to calculate the viral genome copy number per mL.

The RT-qPCR reaction was prepared using the qScript XLT One-Step RT-qPCR ToughMix (QuantaBio, Beverly, MA, USA) in a volume of 12.5 µl including 1 µl of the respective FAM mix and 2.5 µl of extracted RNA. The reaction was performed for 10 min at 50 °C for reverse transcription, 1 min at 95 °C for activation, and 42 cycles of 10 sec at 95 °C for denaturation, 10 sec at 60 °C for annealing and 20 sec at 68 °C for elongation for the Omicron-BA.1-detecting assay. For detection of Alpha and Delta, the following thermal profile was applied: 10 min at 50 °C for reverse transcription, 1 min at 95 °C for activation, and 42 cycles of 5 sec at 95 °C for denaturation, 5 sec at 62 °C for annealing and 10 sec at 68 °C for elongation. Fluorescence was measured during the annealing phase. RT-qPCRs were performed on a BioRad real-time CFX96 detection system (Bio-Rad, Hercules, USA) or with 7500 Fast System SDS (Applied Biosciences, Software Version 1.4). Statistical analysis were done in GraphPad Prism version 8.

## Histopathological and immunohistochemical analysis

The left lung and the left hemisphere of the brain from K18-hACE2 mice were collected upon necropsy and immersed in 10% neutral-buffered formalin. Following fixation, both tissues were embedded in paraffin, cut at 4 µm and stained with hematoxylin and eosin (H&E) for histological evaluation. Lung tissue pathology was scored according to a previously published scoring scheme[10]. A 1:3000 dilution of a rabbit polyclonal anti-SARS-CoV nucleocapsid antibody (Rockland, 200-401-A50) was used for the immunohistochemical

(IHC) analysis of the lung and the brain. Paraffin blocks were cut at 3 μm, placed in a BOND RX™ immunostainer (Leica Byosystems, Germany) and were incubated for 30 min with the first antibody at room temperature. Antigen retrieval was performed by incubating the slides with a citrate buffer for 30 min at 100 °C. Bond™ Polymer Refine Detection visualization kit (Leica Byosystems, Germany) was afterwards used for signal detection using DAB as chromogen and counterstaining with hematoxylin.

### Serological tests
To evaluate the virus neutralizing potential of serum samples, we performed a live virus neutralization test following an established standard protocol as described before[23]. Briefly, sera were prediluted 1/16 in MEM and further diluted in log2 steps until a final tested dilution of 1/4096. Each dilution was evaluated for its potential to prevent 100 TCID$_{50}$ SARS-CoV-2/well of the respective VOC from inducing cytopathic effect in VeroE6 cells, giving the virus neutralization titer (VNT$_{100}$). Additionally, serum samples were tested by multispecies ELISA for sero-reactivity against the SARS-CoV-2 RBD domain[24]. Data is collected with Tecan i-control 2014 1.11 and analyzed with Microsoft Excel 2016 (16.0.5188.1000).

### Human precision-cut lung slice cultures (PCLS)
The generation of PCLSs was done as described previously with some adaptations to human specimens[25]. Control lung tissue (preserved pulmonary architecture without emphysema or inflammation) was obtained from the distal non-tumorous areas of lung resections. Prior to processing, lung tissue was tested for SARS-CoV-2 by qPCR. After gathering, control lung tissue was maintained in DMEM (Thermo Fisher), supplemented with 1X ITS (Sigma) until further processing 2 to 5 h later. Next, lung tissue specimens were washed with PBS (Thermo Fisher) containing 1X Antibiotic-Antimycotic (Thermo Fisher), infused with 2% low-melting point agarose (Sigma) in DMEM, and subsequently put into cold PBS at 4 °C for 15 min to allow the agarose to solidify. Next, the perfused tissue was cut into small cubes of ~1 cm$^3$, placed in the specimen tube and embedded in 2% low-melting point agarose. To generate PCLSs with a thickness of 400 μm, an automated Compresstome VF-310-0Z Vibrating Microtome (Precisionary) was used following the recommended parameters: speed of 8 mm/sec and oscillation of 27 Hz. The slices were transferred into a 12-well plate (one PCLS per well) with culture medium (DMEM, supplemented with 1% FBS, 100 units/mL of penicillin and 100 μg/mL streptomycin, and 2.5 μg/mL of Amphotericin B (all from Thermo Fisher)). Cultures were maintained at 37 °C, 5% CO$_2$ and culture medium changed every 24 h for 2–3 days prior infection.

### Infection of human precision-cut lung slice cultures
PCLS cultures were infected with a 1:1 mixture of SARS-CoV-2$^{S-Delta}$ and SARS-CoV-2$^{S-Omicron}$ in 0.5 mL DMEM, supplemented with 0.1% FBS, 100 units/mL of penicillin and 100 μg/mL streptomycin, and 2.5 μg/mL of Amphotericin B for 2–4 h. Next, the inoculum was removed, PCLSs were washed twice with pre-warmed PBS, and 2 mL of culture medium were added per well. Medium was changed after 24 h. Fourty eight hpi, PCLSs were washed and transferred into cold TRIzol reagent (Thermo Fisher) and kept at −70 °C until further processing.

### RNA isolation of human PCLS
Total RNA was extracted from PCLS cultures using TRIzol reagent in combination with the RNA Clean & Concentrator Kit (Zymo Research). Briefly, Tissue slices were homogenized using MagNA Lyser Green Beads (Roche diagnostics) in combination with a tissue homogenizer (MP Biomedicals) and lysed with 700 μl of cold TRIzol reagent per PCLS. Two hundred mL of chloroform was added to the TRIzol lysate, the samples were mixed vigorously, and then incubated for 2–3 min at room temperature. Next, the extractions were centrifuged at 12,000 g

for 15 min at 4 °C. The aqueous phase was then collected, mixed 1:1 with 75% ethanol, and incubated for 10 min at room temperature to let RNA precipitate. The RNA precipitate was further purified with the RNA Clean & Concentrator Kit according to the manufacturer's instructions.

### Well-differentiated primary nasal and bronchial epithelial cells
Primary human nasal epithelial cell cultures (hNECs) were obtained commercially (Epithelix Sàrl) and primary human bronchial epithelial cell cultures (hBECs) were isolated from lung slices. The generation of well-differentiated (WD)-hNECs and WD-hBECs at the air-liquid interface (ALI) was described previously with minor adjustments[26,27]. For expansion, hNECs and hBECs were cultured in collagen-coated (Sigma) cell culture flasks (Costar) in PneumaCult Ex Plus medium, supplemented with 1 μM hydrocortisone, 5 μM Y-27632 (Stem Cell Technologies), 1 μM A-83-01 (Tocris), 3 μM isoproterenol (abcam), and 100 μg/mL primocin (Invivogen) and maintained in a humidified atmosphere at 37 °C, 5% CO$_2$. Next, the expanded cells were seeded at a density of 50,000 cells per insert onto collagen-coated (Sigma) 24-well plate inserts with a pore size of 0.4 μm (Greiner Bio-One) and grown under submerged conditions with 200 μl of supplemented PneumaCult ExPlus medium on the apical side and 500 μl in the basolateral chamber. When cells reached confluence, as assessed by measuring the trans-epithelial electrical resistance (TEER) using a Volt/Ohm Meter (EVOM$^2$/STX2, World Precision Instruments) and microscopical evaluation, the apical medium was removed, cells were washed with pre-warmed Hank's balanced salt solution (HBSS, Thermo Fisher), and then exposed to the air. PneumaCult ALI medium supplemented with 4 μg/mL heparin (Stem Cell Technologies), 5 μM hydrocortisone, and 100 μg/mL primocin was added to the basolateral chamber to induce differentiation of the cells. Every 2–3 days, the basal medium was changed and cultures maintained at 37 °C, 5% CO$_2$ until the appearance of ciliated cells and mucus production. The cell layer was washed once a week with 250 μL of pre-warmed HBSS for 20 min at 37 °C to get rid of mucus. The hNEC and hBEC cultures were considered well-differentiated 3 weeks post-exposure to ALI.

### Statistical analysis
Statistical analysis was performed using GraphPad Prism 8. Unless noted otherwise, the results are expressed as mean ± s.d. Specific tests are indicated in the main text or the figure legends.

### Figures
Cartoon figures for animal experimentations were created with BioRender.com.

## Data availability
All data are available in the main text or the supplementary materials. Source data are provided with this paper. The project information is accessible with the BioProject ID PRJNA868423. Source data are provided with this paper.

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

## Acknowledgements

For animal care, we thank Frank Klipp, Doreen Fiedler, Christian Lipinski, Harald Manthei and Steffen Kiepert (Friedrich-Loeffler-Institut); I. Wymann, R. Troxler, K. Sliz, and D. Brechbühl (Institute of Virology and Immunology). For technical assistance, we thank Mareen Grawe, Christian Korthase, Patrick Zitzow, Silvia Schuparis, and Bianka Hillmann (Friedrich-Loeffler-Institut). We thank Matthias Lenk for preparing and providing cell cultures and media; and Jacqueline King, Anne Pohlmann, and Lina Stacker for sequencing assistance (Friedrich-Loeffler-Institut). We thank Dr. H.Y. Stoller-Kwan and Dr F. Gregorini, Inselspital, University Hospital Bern, for reagents. We thank Artur Summerfield (Institute of Virology and Immunology) for internal financial support. We thank Georg Kochs from the University of Freiburg, Germany for the SARS-CoV-2 isolate. We thank Franziska. Suter-Riniker and Pascal. Bittel, Institute for Infectious Diseases, University of Bern, for providing clinical samples. VeroE6/TMPRSS2 cells were provided by the NIBSC Research Reagent Repository, UK with thanks to Dr Makoto Takeda. This work was supported by the Swiss National Science Foundation (SNSF) grants no. 31CA30_196062 (CB, RD), 31CA30_196644 (VT, RD), 310030_173085 (VT), 310030_179260 (RD); the Horizon 2020 project "SCORE", grant agreement no. 101003627 (VT); the European Commission, Marie Skłodowska-Curie Innovative Training Network 'HONOURS', grant agreement no. 721367 (VT, RD); Core funds of the University of Bern (VT, RD); Core funds of the German Federal Ministry of Food and Agriculture (MB); the Deutsche Forschungsgemeinschaft (DFG), Project no. 453012513 (MB); the Horizon 2020 project "VEO", grant agreement no. 874735 (MB), the Lungenliga Bern, Switzerland (MPA). The funders had no role in study design, data collection and analysis, decision to publish, or preparation of the manuscript. The authors would like to thank all study participants and their families.

## Author contributions

G.T.B., N.J.H., A.T., J.N.K., J.S., N.E., L.U., R.D., D.H., M.P.A., C.B., M.Be., V.T. conceived the study; G.T.B., N.J.H., A.T., J.S., N.E., L.U., C.D., S.S., B.S.T., I.B.V., D.H. performed most of the experiments; B.H., N.G.F.L., E.A.M., A.B., C.W., D.H., K.W., A.G., L.T., V.F., H.S., M.Br., B.I.O.E., B.Z., G.B., A.K., K.S., S.O., R.M.L., Man.W., C.M., P.D., T.M.M., M.F.-C., A.R., Mar.W., S.C. did experimental work and/or provided essential experimental systems and reagents; J.N.K., C.W. performed sequencing including computational analyses; G.T.B., N.J.H., A.T., J.N.K., J.S., N.E., L.U., D.H., M.P.A., C.B., M.Be., V.T. wrote the manuscript and made the figures. All authors read and approved the final manuscript.

## Competing interests

Authors declare that they have no competing interests.

## Additional information

G. Tuba Barut [1,2,15], Nico Joel Halwe [3,15], Adriano Taddeo [1,2,15], Jenna N. Kelly [1,2,4,5,15], Jacob Schön[3], Nadine Ebert[1,2], Lorenz Ulrich [3], Christelle Devisme[1,2], Silvio Steiner [1,2], Bettina Salome Trüeb[1,2], Bernd Hoffmann [3], Inês Berenguer Veiga [1,2], Nathan Georges François Leborgne[1,2], Etori Aguiar Moreira[1,2], Angele Breithaupt [6], Claudia Wylezich [3], Dirk Höper [3], Kerstin Wernike [3], Aurélie Godel[1,2], Lisa Thomann [1,2], Vera Flück[1,2], Hanspeter Stalder [1,2], Melanie Brügger[1,2], Blandina I. Oliveira Esteves[1,2], Beatrice Zumkehr[1,2], Guillaume Beilleau [1,2,7], Annika Kratzel [1,2], Kimberly Schmied [1,2], Sarah Ochsenbein[1,2], Reto M. Lang[1,2,7], Manon Wider[8], Carlos Machahua[9,10], Patrick Dorn[11,12], Thomas M. Marti [11,12], Manuela Funke-Chambour[9,10], Andri Rauch [4,13], Marek Widera [14], Sandra Ciesek [14], Ronald Dijkman [4,5,8], Donata Hoffmann [3], Marco P. Alves [1,2,4,16] ✉, Charaf Benarafa [1,2,4,16] ✉, Martin Beer [3,5,16] ✉ & Volker Thiel [1,2,4,5,16] ✉

[1]Institute of Virology and Immunology, Bern and Mittelhäusern, Bern, Switzerland. [2]Department of Infectious Diseases and Pathobiology, Vetsuisse Faculty, University of Bern, Bern, Switzerland. [3]Institute of Diagnostic Virology, Friedrich-Loeffler-Institut, Greifswald-Insel Riems, Greifswald, Germany. [4]Multidisciplinary Center for Infectious Diseases, University of Bern, Bern, Switzerland. [5]European Virus Bioinformatics Center, Jena, Germany. [6]Department of Experimental Animal Facilities and Biorisk Management, Friedrich-Loeffler-Institut, Greifswald-Insel Riems, Greifswald, Germany. [7]Graduate School for Cellular and Biomedical Sciences, University of Bern, Bern, Switzerland. [8]Institute for Infectious Diseases, University of Bern, Bern, Switzerland. [9]Department of Pulmonary Medicine, Inselspital, Bern University Hospital, University of Bern, Bern, Switzerland. [10]Department for Pulmonary Medicine, BioMedical Research, University of Bern, Bern, Switzerland. [11]Division of General Thoracic Surgery, Inselspital, Bern University Hospital, University of Bern, Bern, Switzerland. [12]Department for BioMedical Research, Inselspital, Bern University Hospital, University of Bern, Bern, Switzerland. [13]Department of Infectious Diseases, Inselspital, Bern University Hospital, University of Bern, Bern, Switzerland. [14]Institute of Medical Virology, University Hospital Frankfurt, Goethe University, Frankfurt am Main, Germany. [15]These authors contributed equally: G. Tuba Barut, Nico Joel Halwe, Adriano Taddeo, Jenna N. Kelly. [21]These authors jointly supervised this work: Marco P. Alves, Charaf Benarafa, Martin Beer, Volker Thiel. ✉e-mail: marco.alves@vetsuisse.unibe.ch; charaf.benarafa@vetsuisse.unibe.ch; martin.beer@fli.de; volker.thiel@vetsuisse.unibe.ch

