## [Peer Review File · Nature Communications]

The spike gene is a major determinant for the SARS-CoV-2 Omicron-BA.1 phenotypeREVIEWERS' COMMENTS

Reviewer #1 (Remarks to the Author):

This work performed characterization of Omicron-BA.1 and recombinant Omicron-BA.1 spike gene mutants comparing to the Delta VOC in well-differentiated primary human nasal and bronchial epithelial cells in vitro followed by in vivo fitness characterization in naïve hamsters, ferrets and hACE2-expressing mice and in immunized hACE2-mice.

The results indicate that the spike gene is a major determinant of Delta and Omicron-BA.1 replication and pathogenicity. Moreover, it seems that Omicron-BA.1 dominance may be also explained by spike gene-mediated immune evasion as an additional factor.

It is well described the list of mutations in Omicron-BA.1 including about 34 substitutions in the spike (S) gene, 15 of them in the Receptor-Binding Domain (RBD). This VOC has a remarkable ability to evade neutralizing antibodies up to 40 times more efficiently than the ancestral SARS-CoV-2 and pre-Omicron variants, an ability also present among double-vaccinated and even boosted individuals.

The authors has obtained some important results, including:

1. Replication in nasal but not bronchial epithelium enhanced by Omicron-BA.1 spike: this conclusion was possible due to authors constructed recombinant SARS-CoV-2 clones with defined mutations on the spike gene, including specific mutations for Delta, Omicron-BA.1, in NTD of Omicron-BA.1, RDB Omicron-BA.1, and mutations on spike cleavage site region of Omicron-BA.1.
2. Replicative fitness and transmission of Alpha, Delta and Omicron in Syrian hamsters: the results indicated the animals were susceptible for all VOCs, but Alpha seems to better replicate and transmit on those animals compared to Delta and Omicron. When Delta vs Omicron was tested in hamsters, Delta was immediately prevalent in nasal washings, which is not concordant with human epidemiological situation where Omicron-BA.1 rapidly became prevalent. The authors concluded that this possibly indicates adaptation of SARS-CoV-2 variants to the human host.
3. Omicron does not induce productive infection in ferrets, animals that mimize human respiratory disease: there was a complete replication block of Omicron-BA.1 in ferrets, while Delta exhibits top-level fitness, leading to the conclusion that the higher number of mutations in the Omicron-BA.1 limited the host spectrum and there was adaptation of this variant to human host. Authors suggest that Omicron-BA.1 being outcompeted by Delta in naive Syrian hamsters and completely blocked in ferrets indicates it is unlikely that Omicron variant evolved in an animal reservoir.
4. Mutations in the Delta spike gene enhanced fitness in humanized mice: knock-in mice expressing only human ACE2 (hACE2-KI) inoculated intranasally with Omicron-BA.1 or Delta showed higher viral load and titers in mice inoculated with Delta when compared to Omicron-BA.1, with consequently higher pathological score in lungs. When mice were inoculated with an equivalent mixture of recombinant clones with different mutations in the spike sequence, Delta fully dominated over Omicron, leading to the conclusion that those mutations in Omicron spike do not provide advantage over Delta in humanized mouse models. Consequently, spike mutations seem to be determinant for the replacement of Delta by Omicron-BA.1 in human populations. Authors concluded that the replicative properties of Omicron-BA.1 are not the decisive factor for this replacement.
5. The spike in Omicron-BA.1 confers immune escape and reduced pathogenicity: Authors investigated the vaccination with mRNA using spike sequence as the sole viral antigen and the variants response. Omicron-BA.1 spike gene confers less virulent phenotype than 614G or Delta spike gene and also confers the largest degree of immune evasion since there were comparable infectious titers in vaccinated and unvaccinated mice.

Importantly, the authors emphasize the influence of Omicron-associated immune escape potential and the importance of immune status, including through vaccination, on virus selection.

I consider this work well-designed and well-written, in a synthetic and clear format. Considering an evolutionary perspective, the results are of high interest to the academic community and also to a broader audience including health professionals. The study illustrates and indicates the remarkable evolutionary and phenotypic jump impacting viral replication, pathogenicity, and immune escape. It also considers the impact of vaccination in the viral host adaptation and help us to better understand the evolutionary impact and importance of mutations in the spike protein. In

conclusion, I recommend the article publication considering the correct methodological design and all interesting and important results and conclusions obtained by the authors.

Reviewer #2 (Remarks to the Author):

The study is thorough and provides a useful comparison between Delta and Omicron BA1 that is well executed, described and interpreted. It provides a context on how different VOCs compete with one another

The manuscript is (unfortunately) no longer at the cutting edge of SARS-CoV=2 research as BA.11 and been rapidly replaced with other variants; nevertheless the study provides useful information. However, the text should be modified accordingly (eg Abstract/Introduction)

The finding that BA.1 infection is abortive in ferrets is notable

Reviewer #3 (Remarks to the Author):

The authors of this manuscript describe fitness studies of SARS-CoV-2 attributed to the spike of Omicron BA.1 compared to the spike of Delta both in cell culture and animal experiments. They found that authentic BA.1 and a chimeric virus possessing the BA.1 spike has more fit (replicates better) in upper respiratory tract epithelium cells at 33C compared to bronchial epithelial cells at 37C. In contrast, the Delta variant or a chimeric virus with the Delta spike replicated better in BECs at 37C. The authors found similar results in competition assays where cell lines were co-infected with two different variants or chimeric viruses. The BA.1 variant or a chimeric virus with the BA.1 spike outcompeted Delta or an ancestral S-614G viruses in upper respiratory epithelium cells at 33C whereas this dominance was not observed in BECs at 37C.

In a hamster Direct Contact transmission model, the authors demonstrated that Alpha outcompeted Delta though the animals generated similar neutralizing antibody titers against both variants. And while there was a ~ 1:2 ratio of Delta to BA.1, the Delta variant outcompeted BA.1 both in the donor and the contact hamsters. A similar study was performed with ferrets; however, BA.1 failed to replicate efficiently in these animal model.

To examine competition in an animal model with similar receptor usage, the authors used transgenic human ACE2 mice. Similar to the hamster studies, the Delta variant outcompeted BA.1.

The title of manuscript implies that the Spike of BA.1 is a major determinant of the BA.1 phenotype. This title can be justified by Figure 1 and then again in Figure 4d. However, the authors data conflicts with data on the XD variant in mice which replicates better than BA.1; do the authors discuss these findings?

The use of a chimeric virus in the background of BA.1 with a Delta spike would confirm the authors' findings and should be performed.

Minor points:

Please clarify that the transmission studies were based on direct contact verse airborne transmission.

Removed "humanized" to describe the hACE2 knock-in mice.

RESPONSE TO REVIEWERS' COMMENTS

Authors's response to the reviewers:

We appreciate the time and effort that the editor and the reviewers have dedicated to providing valuable feedback on the manuscript. We are grateful to the reviewers for their insightful comments. We have been able to incorporate the highlighted changes within the manuscript, to reflect most of the suggestions provided by the reviewers. Here is a point-by-point response to the reviewers' comments and concerns:

Reviewer #1 (Remarks to the Author):

This work performed characterization of Omicron-BA.1 and recombinant Omicron-BA.1 spike gene mutants comparing to the Delta VOC in well-differentiated primary human nasal and bronchial epithelial cells in vitro followed by in vivo fitness characterization in naïve hamsters, ferrets and hACE2-expressing mice and in immunized hACE2-mice. The results indicate that the spike gene is a major determinant of Delta and Omicron-BA.1 replication and pathogenicity. Moreover, it seems that Omicron-BA.1 dominance may be also explained by spike gene-mediated immune evasion as an additional factor. It is well described the list of mutations in Omicron-BA.1 including about 34 substitutions in the spike (S) gene, 15 of them in the Receptor-Binding Domain (RBD). This VOC has a remarkable ability to evade neutralizing antibodies up to 40 times more efficiently than the ancestral SARS-CoV-2 and pre-Omicron variants, an ability also present among double-vaccinated and even boosted individuals.

The authors has obtained some important results, including:

1. Replication in nasal but not bronchial epithelium enhanced by Omicron-BA.1 spike: this conclusion was possible due to authors constructed recombinant SARS-CoV-2 clones with defined mutations on the spike gene, including specific mutations for Delta, Omicron-BA.1, in NTD of Omicron-BA.1, RDB Omicron-BA.1, and mutations on spike cleavage site region of Omicron-BA.1.
2. Replicative fitness and transmission of Alpha, Delta and Omicron in Syrian hamsters: the results indicated the animals were susceptible for all VOCs, but Alpha seems to better replicate and transmit on those animals compared to Delta and Omicron. When Delta vs Omicron was tested in hamsters, Delta was immediately prevalent in nasal washings, which is not concordant with human epidemiological situation where Omicron-BA.1 rapidly became prevalent. The authors concluded that this possibly indicates adaptation of SARS-CoV-2 variants to the human host.
3. Omicron does not induce productive infection in ferrets, animals that mimetize human respiratory disease: there was a complete replication block of Omicron-BA.1 in ferrets, while Delta exhibits top-level fitness, leading to the conclusion that the higher number of mutations in the Omicron-BA.1 limited the host spectrum and there was adaptation of this variant to human host. Authors suggest that Omicron-BA.1 being outcompeted by Delta in naive Syrian hamsters and completely blocked in ferrets indicates it is unlikely that Omicron variant evolved in an animal reservoir.

4. Mutations in the Delta spike gene enhanced fitness in humanized mice: knock-in mice expressing only human ACE2 (hACE2-KI) inoculated intranasally with Omicron-BA.1 or Delta showed higher viral load and titers in mice inoculated with Delta when compared to Omicron-BA.1, with consequently higher pathological score in lungs. When mice were inoculated with an equivalent mixture of recombinant clones with different mutations in the spike sequence, Delta fully dominated over Omicron, leading to the conclusion that those mutations in Omicron spike do not provide advantage over Delta in humanized mouse models. Consequently, spike mutations seem to be determinant for the replacement of Delta by Omicron-BA.1 in human populations. Authors concluded that the replicative properties of Omicron-BA.1 are not the decisive factor for this replacement.

5. The spike in Omicron-BA.1 confers immune escape and reduced pathogenicity: Authors investigated the vaccination with mRNA using spike sequence as the sole viral antigen and the variants response. Omicron-BA.1 spike gene confers less virulent phenotype than 614G or Delta spike gene and also confers the largest degree of immune evasion since there were comparable infectious titers in vaccinated and unvaccinated mice.

Importantly, the authors emphasize the influence of Omicron-associated immune escape potential and the importance of immune status, including through vaccination, on virus selection.

I consider this work well-designed and well-written, in a synthetic and clear format. Considering an evolutionary perspective, the results are of high interest to the academic community and also to a broader audience including health professionals. The study illustrates and indicates the remarkable evolutionary and phenotypic jump impacting viral replication, pathogenicity, and immune escape. It also considers the impact of vaccination in the viral host adaptation and help us to better understand the evolutionary impact and importance of mutations in the spike protein. In conclusion, I recommend the article publication considering the correct methodological design and all interesting and important results and conclusions obtained by the authors.

Response to Reviewer #1:

We highly appreciate the reviewer's insightful comments on our manuscript.

Reviewer #2 (Remarks to the Author):

The study is thorough and provides a useful comparison between Delta and Omicron BA1 that is well executed, described and interpreted. It provides a context on how different VOCs compete with one another

The manuscript is (unfortunately) no longer at the cutting edge of SARS-CoV=2 research as BA.11 and been rapidly replaced with other variants; nevertheless the study provides useful information. However, the text should be modified accordingly (eg Abstract/Introduction)

The finding that BA.1 infection is abortive in ferrets is notable.

Response to Reviewer #2:

We are grateful for the reviewer's comments on our manuscript. We have updated the manuscript accordingly (see track changes in the main text).

Reviewer #3 (Remarks to the Author):

The authors of this manuscript describe fitness studies of SARS-CoV-2 attributed to the spike of Omicron BA.1 compared to the spike of Delta both in cell culture and animal experiments. They found that authentic BA.1 and a chimeric virus possessing the BA.1 spike has more fit (replicates better) in upper respiratory tract epithelium cells at 33C compared to bronchial epithelial cells at 37C. In contrast, the Delta variant or a chimeric virus with the Delta spike replicated better in BECs at 37C. The authors found similar results in competition assays where cell lines were co-infected with two different variants or chimeric viruses. The BA.1 variant or a chimeric virus with the BA.1 spike outcompeted Delta or an ancestral S-614G viruses in upper respiratory epithelium cells at 33C whereas this dominance was not observed in BECs at 37C.

In a hamster Direct Contact transmission model, the authors demonstrated that Alpha outcompeted Delta though the animals generated similar neutralizing antibody titers against both variants. And while there was a ~ 1:2 ratio of Delta to BA.1, the Delta variant outcompeted BA.1 both in the donor and the contact hamsters. A similar study was performed with ferrets; however, BA.1 failed to replicate efficiently in these animal model. To examine competition in an animal model with similar receptor usage, the authors used transgenic human ACE2 mice. Similar to the hamster studies, the Delta variant outcompeted BA.1.

The title of manuscript implies that the Spike of BA.1 is a major determinant of the BA.1 phenotype. This title can be justified by Figure 1 and then again in Figure 4d. However, the authors data conflicts with data on the XD variant in mice which replicates better than BA.1; do the authors discuss these findings?

The use of a chimeric virus in the background of BA.1 with a Delta spike would confirm the authors' findings and should be performed.

Minor points:

Please clarify that the transmission studies were based on direct contact versus airborne transmission.

Removed “humanized” to describe the hACE2 knock-in mice.

Response to Reviewer #3:

We thank the reviewer for the constructive comments.

- 1) The reviewer refers to published data on the XD variant which should replicate better than BA.1 and asked to discuss these data.

We found relevant information only in a preprint publication (Simon-Lorieri et al., <https://doi.org/10.21203/rs.3.rs-1502293/v1>) on in vitro and in vivo characterization of a Delta-Omicron recombinant virus. In this study, authors characterize a Delta-Omicron recombinant virus that carries a larger fraction of the signature changes of BA.1 spike, including the entire receptor binding domain.

In line with our findings and literature, their experiments with K18- hACE2 mice show that when infected with Delta, there are higher infectious virus titres in lungs and nasal turbinates at 3 dpi, compared to the mice infected with BA.1. When they compare the recombinant virus that has Delta backbone + BA.1 spike to the other viruses, they detect comparable levels of XD and BA.1 in the lungs (lower than Delta), but similar levels of XD and Delta in the nasal turbinated that were higher than BA.1. This suggests that the XD virus acts similar to BA.1 in the lungs and similar to Delta in the nose.

We think that Reviewer 3 might have been indicating the finding that BA.1 replicating more in the nasal turbinates than XD is conflicting with our data. However, BA.1 spike combined to Delta was not tested in the experiments we performed. Viruses with a syngenic backbone were used to exclusively attribute phenotypes to related spike sequences. We agree with Reviewer 3 and the authors of the preprint on the notion that we cannot rule out the role of the viral backbone in the course of infection. However, we would like to argue that the scope of our study is to reveal the importance of the contribution of the spike during the course of infection, and how much difference it makes both in vitro and in vivo. The recombination of two or more variants, on the other hand, is a very compelling phenomenon and needs to be further investigated. Unfortunately, it was not possible to fit more recombinant viruses that are currently in circulation around the world into our timeline and sample sizes. We fully understand the point of Reviewer 3, and would very much appreciate their understanding why doing these various new combinations of different recombinant viruses was not be feasible in the context of this study.

- 2) To corroborate our findings the reviewer is asking for a reverse experiment with a Delta spike in the background of BA.1.

We thank the reviewer for the qualified comment. The reviewer has commented that additional experiments with a chimeric virus in the background of BA.1 and a Delta spike would confirm our findings. Although we totally agree with the reviewer, considering the huge amount of work put in this paper, as well as the amount of work and time another set of these experiments would require, we would like to keep the question of impact of the virus backbone for further upcoming studies.

Minor points:

- 3) The reviewer advises to clarify that the inoculated and naïve contact animals were kept in direct contact and not in an airborne-transmission-only setup.

In the methods parts it is described that donor and contact animals were co-housed 24 hours after inoculation of the donor animals. The phrase “allowing direct contact of donor to contact ferrets” was written for the ferret experiments and is now also written in the method section for the hamster experiments.

- 4) The reviewer suggested to remove the term “humanized” when it comes to describe hACE2-K18-mice.

We agree that this term is not exact to describe mice which express in addition to the mice ACE2 receptor, the human ACE2 receptor. As suggested by the reviewer, we removed the word “humanized” from the text.

Additional changes applied to the manuscript:

Main text

All “lung explants” in the text are corrected to “precision-cut lung slices”.

Abstract

Lines 50-68: Abstract is shortened to 200 words, and the text is updated to present tense.

Methods

Lines 378-383: A biosafety statement is added in the Methods section.

It reads: “**Biosafety Statement**

All experiments with infectious SARS-CoV-2 were performed in enhanced biosafety level 3 (BSL3) containment laboratories at Institute of Virology and Immunology, Mittelhäusern, Switzerland, and Friedrich-Loeffler-Institut, Greifswald-Insel Riems, Germany, which followed the approved standard operating procedures of BSL3 facility of relevant authorities

in Switzerland and Germany. Before commencing work, all personnel received relevant training.”

Lines 627-632: Mouse studies:

A new paragraph is added: “Mice were produced at the specific-pathogen-free facility of the Institute of Virology and Immunology (Mittelhäusern), where they were maintained in individually ventilated cages (blue line, Tecniplast), with 12-h/12-h light/dark cycle, 22 ± 1 °C ambient temperature and $50 \pm 5\%$ humidity, autoclaved food and acidified water. At least 7 days before infection, mice were placed in individually HEPA-filtered cages (IsoCage N, Tecniplast).”

Lines 778-779: New section is added.

Figures: Cartoon figures for animal experimentations were created with BioRender.com.

Data and Material availability:

Lines 774-777: The text “The project information is accessible with the BioProject ID PRJNA868423 (<https://www.ncbi.nlm.nih.gov/bioproject/?term=PRJNA868423>).” is added to the Data and Material availability section.

Figures

Figure 1: p-values are added on the graphs.

Figure 3: p-values are added on the graphs.

Figure 4: p-values are added on the graphs. “%Body weight change” axis title is added to Fig4b. “Oropharyngeal swabs, nose, lung” graph titles are added to the lower panels of Fig4c and d.

Supplementary Figure 10: p-values are added on the graphs. A new graph showing the body weight change of individual mice is added as Supp.Fig.10a. The other graphs have shifted in the alphabetical order.